# P2PRISM - Peer to peer learning with individual prism for model aggregation

## Abstract

Federated learning (FL) has made collaboration between nodes possible without explicit sharing of local data. However, it requires the participating nodes to trust the server and its model updates, the server itself being a critical node susceptible to failure and compromise. A loss of trust in the server and a demand to aggregate the model independently for oneself has led decentralized peer-to-peer learning (P2PL) to gain traction lately. In this paper, we highlight the never before exposed vulnerabilities of P2PL towards malicious attacks and show the differences between the behavior of P2PL and FL in a malicious environment. We then present a robust defense - P2PRISM as a secure aggregation protocol for P2PL.

## 1 Introduction

### 1.1 Motivation for peer-to-peer learning

FL McMahan et al. (2017); Konečný et al. (2016) has demonstrated how clients can benefit from collaboration by sharing their local gradient updates to the parameter server, which in turn aggregates Yin et al. (2018a); Blanchard et al. (2017); Guerraoui et al. (2018); Xia et al. (2019); Fung et al. (2020); Cao et al. (2020) all the received gradients to update the global model. For the sake of simplicity, we assume that the entire process is synchronous - the server waits to hear from all the clients before aggregation and all clients receive the same global model from the server after aggregation, that is, the clients always have to agree on the global model sent by the server and replace its local model with it before continuing with the local training. Although the aggregation technique being used may be known to all, but the actual aggregation is hidden from the clients for privacy concerns as it has been shown that access to a client's gradients can be used to recover its local data in an approximate or an exact way by optimization Geiping et al. (2020) or analytical Fowl et al. (2021) methods respectively. It is therefore not possible for clients to selectively choose other clients' gradients to aggregate even if it benefits them from any existing spatial locality among the clients. The clients have to trust the server to also aggregate the gradients in a byzantine-robust manner. Unless the server itself possesses a root dataset Cao et al. (2021) that correctly represents the entirety of data possessed by all clients as the ground truth, it is difficult for it to correctly identify malicious updates statistically without being extremely conservative and removing any suspected gradients leading to a significant loss of information. Whereas a node does have access to its own generated gradients as the benign ground truth and can make use of it, given the power to aggregate the model for itself. Due to the above mentioned reasons, and several others, a node is motivated to lose trust in a server and join a decentralized collaboration among the other nodes.

### 1.2 Comparison with Federated learning

In FL, the server aggregates the gradients from all the clients. However, in P2PL, a node may choose to communicate only with its neighbors in every round and locally aggregate the received models. If the graph formed by the nodes is not fully connected with equally weighed edges, the nodes are going to have models that differ from each other at every point in time even after their local aggregation. Consensus distance ($\delta$) of the graph is defined as the average distance of each of the $m$ local models ($x_i$) from the centroid ($\bar{x}$) of them all, known to an oracle.

$$\delta := \frac{1}{m} \sum_{i=1}^{i=m} \|x_i - \bar{x}\|$$ (1)

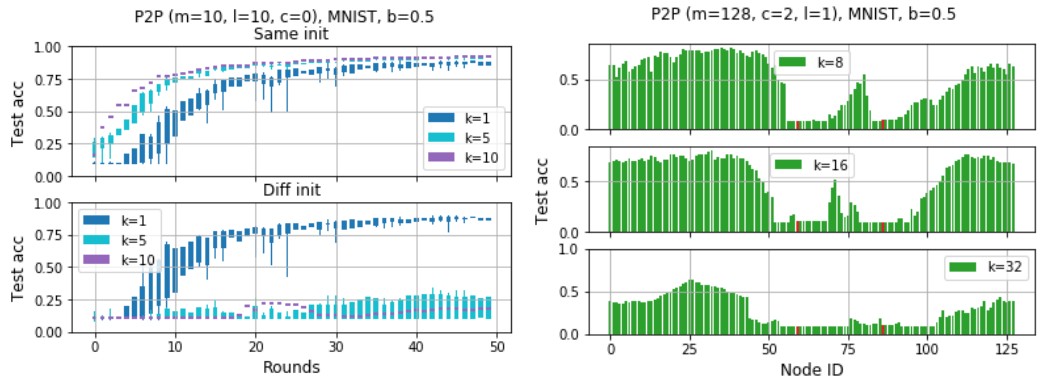

Figure 1: *Left*: Training a k-regular peer-to-peer graph with varying values of $k$ with same and different model initialization. We see that it is necessary to initialize nodes with the same model to expect benefit from collaboration ($k > 1$). *Right*: Demonstrating the spread of the attack in a k-regular graph where only 2 out of 128 nodes (shown in red) are malicious. We can see that the larger the collaboration ($k$), the greater the spread of the attack if the aggregation used is insecure, such as gossip averaging in this case.

It is measured after every aggregation step. Needless to say this takes a zero value in FL and non-zero value in P2PL when the graph is not fully connected. A non-zero consensus value implies that sharing gradients as in FL is not appropriate in P2PL and the nodes need to share the actual model weights with their neighbors because the local gradients have been computed on already differing local models. The nodes, after performing local SGD on their own data, where the nodes' data can be assumed to be non-IID with bias $= b$ to incentivize collaboration, tend to diverge from each other. In FL, the server enforces full consensus among the clients, whereas in P2PL, the nodes perform one or more rounds of gossip averaging (GA) with their neighbors to keep the consensus distance in check. It is always wise to initialize all nodes in P2PL with the same model, that is, zero initial consensus distance to aid consensus control in the later stages of training, as also demonstrated in the toy experiment in Figure 1. We see that with same initialization, the nodes benefit from collaboration as $k$ increases to 5 and 10 from 1 (individual training). This is also why every client in FL benefits from collaboration as their models are made to synchronize by the server. Whereas, when nodes are initialized differently, they hurt each other with collaboration, because they are all on different trajectories towards learning an optimal model, as the same ML problem may have multiple solutions depending on the initialization. In fact, we observe that the nodes are hurt even with $k = m = 10$ when after a single round of gossip, the consensus distance falls down to 0 and stays at 0. Hence, keeping the pre-gossip consensus distance low is as important as keeping the post-gossip consensus distance, which is why we recommend setting the initial pre-gossip consensus distance to 0 by initializing all the nodes with the same model.

Assuming a weighted mean aggregation, in FL, the server assigns weights to all the clients. In P2PL, every node is responsible for its own aggregation, and maintains a vector of weights (zero or non-zero) it assigns to every other node, including itself. These vectors when stacked on top of each other form the mixing matrix $W$ of the graph and define its topology. If $x$ is a matrix with $x_i$ being the model of the $i^{th}$ node, then the gossip averaging step is captured by the matrix operation $Wx$. It is to be noted that a node is only affected by its direct neighbors in one round of gossip averaging, but can be affected by an indirectly connected neighbor if multiple rounds are performed. For example two rounds of GA with matrix $W$ results in the models $W \cdot Wx$ which is effectively a single round of GA with matrix $W^2$ where $W^2$ is less sparse than $W$ for positive entries in $W$. It should also be obvious that more the number of gossip averaging steps, the tighter the consensus control Kong et al. (2021).

## 1.3 P2PL IN A BYZANTINE ENVIRONMENT

P2PL behaves differently from FL under a malicious setting. Assuming the same model poisoning attack in both the cases, with $c$ out of $m$ nodes being malicious or compromised, while the FL server has to deal with $f(= \frac{c}{m})$ fraction of malicious nodes, the nodes in P2PL have to deal with

a variable fraction of malicious neighbors which could be greater than $f$ for many, depending on the distribution and connectivity of the nodes. Attackers can not only impact their direct neighbors, but the impact spreads like a disease with either multiple GA steps in the same round, or even with a single GA step when other nodes come in contact with the poisoned nodes in the next round before the infected could recover themselves during their local SGD. In this manner, the attacker also succeeds in magnifying the consensus distance among the nodes that makes training even more difficult. Depending on the graph topology and distribution of the attackers, although an attacker may not be able to impact every node as in FL, the impact on affected nodes can be higher because some nodes are bound to have $\geq f$ fraction of their neighbors malicious. Figure 1 demonstrates this phenomenon in P2PL in comparison with FL.

### 1.4 THREAT MODEL

We have extended the state-of-the-art model poisoning attack, SHEJWALKAR attack Shejwalkar & Houmansadr (2021) in FL to the P2PL case. The malicious nodes collaborate among themselves to access the complete knowledge of the current as well as the past state of the benign models. A perturbation unit vector is constructed along the direction in which the average benign model is moving, and would be scaled up and subtracted from the past averaged model to send the nodes along the direction of gradient ascent. The scaling factor is chosen by solving an optimizing function that balances between attack impact and stealth, given that we know the aggregation technique being used. The attack thus amplifies the distance between an infected and an uninfected node, leading them to disagreement and thereby breaking consensus among the nodes.

### 1.5 OUR DEFENSE - P2PRISM

Our defense is based on the key intuition that the direction of the received models should be of utmost importance during aggregation. This is difficult in FL as there is no way for the server to know which clients are benign to use as reference to compute the direction of the updates. Naturally, none of the aggregation techniques that are expected to be byzantine-robust in FL make use of this fact. In P2PL, a node can use its own local update as the reference to detect parameter-wise direction flip in the updates received from its neighbors. It can then filter out suspected updates based on a defence policy that we call as $PRISM$ before aggregating the received updates. We describe this policy in much detail in the next section, and we use it to completely undo the damage done by the attack, and reviving upto 100% of the benign nodes in a graybox attack and upto 88% of them in a k-regular graph in the presence of a whitebox attack.

In summary, our contributions are -

- We demonstrate the effect of byzantine nodes in P2PL and propose a defense against the state-of-the-art attack by leveraging the ground truth information available in P2PL as opposed to FL, reviving upto 100% of the nodes from the effect of the attack.

- We explore the behavior of P2PL across k-regular and power-law graph topologies and highlight the subtleties involved in training like model initialization that affect consensus among the nodes. We also empirically show how P2PRISM keeps the consensus distance under control.

- We evaluate P2PRISM on the image datasets of MNIST and FMNIST, and the NLP dataset of Shakespeare under graybox as well as fully whitebox adaptive attacker capabilities to prove effectiveness of our defense principle under the ultimate stress-test environment, where upto 88% of the nodes in a k-regular graph and 66% nodes in a power-law graph were completely revived even when 1/4-th of the nodes were under the control of a whitebox attacker.

## 2 DESIGN

P2PRISM is instantiated at every node to perform secure aggregation. It provides the node with a prism that blocks the passage of malicious updates into the aggregated model. P2PRISM uses a metric called flip-score Sharma et al. (2021) (FS) to detect malicious updates and helps a node keep a record of the reputation score of all its neighbors. The flip-score is expected to capture any large deflection as compared to a trusted benign update. A node $i$ with model $x_{i,t}$ sets $\nabla x_{ii} = x_{i,t} - x_{i,t-1}$

as the trusted reference. All models that is receives from its neighbors $j$ are adjusted according to this reference: $\nabla x_{ij} = x_{j,t} - x_{i,t-1}$. The flip-score of $j$ as computed by $i$ is

$$FS_{ij} = \sum_{k=0}^{|P|-1} (|\nabla x_{ij}[k]|)^2 \times (sign(\nabla x_{ij}[k]) \neq (sign(\nabla x_{ii}[k]))). \tag{2}$$

where $|P|$ is the total number of parameters (weights and biases) in the model. Since a node can trust its own update $\nabla x_{ii}$ to go towards gradient descent, a high flip-score naturally captures an update that is likely going towards a gradient ascent - with either a large number of parameters going in the opposite direction with a small magnitude or a small number of them with a large magnitude. Cosine similarity is also based on a similar intuition but it does not capture a subtle attack that targets model parameters with zero gradients to prevent them from converging. With this, we can say that an abnormally large FS is indicative of an attack but it is still not enough by itself without a defense policy.

In FL, byzantine-robust aggregation technqiues such as Trimmed Mean and Median Yin et al. (2018b), Krum Blanchard et al. (2017), Bulyan Mhamdi et al. (2018), FABA Xia et al. (2019), FLTrust Cao et al. (2021) and others assume a constant upper limit on the number of malicious nodes ($c_{max} > \frac{m}{2}$). That may work when the attackers are not in a majority, by trimming out a fixed number of updates in every iteration and penalizing the reputation of the respective clients. The attackers may, however form a local majority of variable size at different regions in P2PL, and complete security cannot be maintained with a constant $c_{max}$. We describe how P2PRISM handles this in the following steps.

1. **Finding the cutoff FS** - In P2PRISM, every node finds the minimum ($FS_{min}$) and median FS ($FS_{med}$) for all the updates it has received in a given iteration. It then sets a cutoff FS at $FS_{med} + \mu(FS_{med} - FS_{min})$, where $\mu \geq 0$ is our only defense parameter. A lower $\mu$ sets a conservative policy whereas a large value sets a lenient one. For a robust computation, it is suggested to use a lower percentile instead of the absolute minimum, when the number of neighbors is sufficiently large ($\geq 100$ for example).

2. **Reward or penalize neighbors** - All neighbors with FS less than the cutoff are given a constant reward of 1 unit, which prevents a node from accumulating high rewards as compared to others. All others are penalized by the variable amount $max(\frac{FS-FS_{med}}{FS_{med}-FS_{min}}, 5)$ so that those close to but higher than the median are penalized less than those far from and higher than the same. The penalty is upper capped so that a node is allowed to redeem itself if it acts benign in the future.

3. **Update reputation** - The reward or penalty is added to the previous reputation score that was initialized to $\frac{1}{k}$ for a neighbor of a node that has $k-1$ neighbors. In this way, we make use of the past behavior of every neighbor. The reputation scores are normalized by dividing each value with the total positive reputation score of all neighboring nodes. Avoiding negative weights for normalization prevents the denominator from getting too small. All neighbors with a negative reputation are filtered out by this defense policy. .

4. **Aggregation** - The remaining updates, including a node's self update, undergo mean aggregation. A weighted mean weighed by the reputation is intentionally avoided to prevent the effect of any existing bias. For example. Nodes with similar updates could assign each other a high weight and isolate themselves from the rest of the graph, and not benefit from collaboration to learn a generalized model. This would also result in a higher consensus distance among the nodes.

In order to protect a node surrounded by malicious majority, we also cap its $FS_{med}$ to be $\leq 10 * FS_{min}$. Without this, the median value could be poisoned leading to a victory of the attackers. The pseudocode of P2PRISM is described in Algorithm 1. P2PRISM is instantiated at every node in the peer-to-peer graph. It is to be noted that we use FS only to trim out the anomalies, that affects the mixing matrix weights in every round. Hence the convergence analysis follows from already existing work Assran et al. (2019) that proves convergence for changing topologies under column stochasticity condition. We do not assume a doubly stochastic mixing matrix as in Koloskova et al. (2020) or column stochastic as in Assran et al. (2019) but for ideal security want all entries in the column for a malicious node to be 0. Hence, we only assume it to be row stochastic. The rest of the assumptions are standard as follows -

**Assumption #1**: (L-smoothness). We assume that for models $x, y in \mathbb{R}^d$, and local gradients $\nabla f_i(\cdot)$ of node $i$, there exists a constant $L > 0$ such that $\|\nabla f_i(y) - \nabla f_i(x)\| \leq L \|y - x\|$.

---

**Algorithm 1** Peer-to-peer learning with P2PRISM

---

**Input**: Node $i$, current model $x_i(t, \cdot)$,
         Neighbor updates $x_j(t+1, \cdot)$, neighbors' reputation $W_R(t, j)$
**Output**: Aggregated model $x_i(t+1, \cdot)$
**Parameters**: $\mu$

**0 :** Compute reference vector $\nabla x_{ij} = x_i(t+1, \cdot) - x_i(t, \cdot)$
**1 :** for every neighbor $j$ **compute flip-score**:
     $FS_{ij} = \sum_{k=0}^{|P|-1} (|\nabla x_{ij}[k]|)^2 \times (sign(\nabla x_{ij}[k]) \neq (sign(\nabla x_{ii}[k]))$
**2 :** Compute cutoff flipscore:
     $FS_{min,i} = min_j\{FS_{ij}\}$
     $FS_{med,i} = min(median_j\{FS_{ij}\}, 10 \times FS_{min,i})$
     $FS_{i,cut} = FS_{med,i} + \mu \times (FS_{med,i} - FS_{min,i})$
**3 : Penalize** neighbors with high FS:
     $W_R(t+1, j) = W_R(t, j) - max\{\frac{FS_{ij} - FS_{med,i}}{FS_{med,i} - FS_{min,i}}, 5\}$
**4 : Reward** the rest of the neighbors:
     $W_R(t+1, j) = W_R(t, j) + 1$
**5 : Normalize** reputation weights: $W_R(t+1, j) = \frac{W_R(t+1,j)}{\sum_{j:W_R(t+1,j)>0} W_R(t+1,j)}$
**6 : Aggregate** gradients: $x_i(t+1, \cdot) = x_i(t, \cdot) + avg_{j:W_R(t+1,j)>0}(x_j(t+1, \cdot) - x_i(t, \cdot))$

---

**Assumption #2:** (Bounded variance). Let $\xi_t^j$ be sampled uniformly from the local data $D_j$ of the $j - th$ node, then the variance of stochastic gradients of each client is bounded, that is, $\mathbb{E}_{\xi \sim D_i} \|\nabla F_i(\boldsymbol{x}; \xi) - \nabla f_i(\boldsymbol{x})\|^2 \leq \sigma^2, \quad \forall i, \forall \boldsymbol{x}$ , and the variance across nodes is also bounded, $\frac{1}{m} \sum_{i=1}^m \|\nabla f_i(\boldsymbol{x}) - \nabla f(\boldsymbol{x})\|^2 \leq \zeta^2 \forall \boldsymbol{x}$.

**Assumption #3:** (Mixing connectivity) To each mixing matrix $\mathbf{P}^{(k)}$ at iteration $k$, we can associate a graph with vertex set $\{1, \ldots, m\}$ and edge set $E^{(k)} = \{(i, j) : w_{i.j}^{(k)} > 0\}$. We assume that there exists finite, positive integers, $B$ and $\Delta$, such that the graph with edge set $\bigcup_{k=lB}^{(l+1)B-1} E^{(k)}$ is strongly connected and has diameter at most $\Delta$ for every $l \geq 0$.

We show that for for $K$ greater than a finite limit, $\overline{x}^{(k)} = \frac{1}{m} \sum_{i=1}^m x_i^{(k)}$ converges, that is, $\frac{1}{K} \sum_{k=1}^K \mathbb{E} \left\| \nabla f \left( \overline{x}^{(k)} \right) \right\|^2 \leq \epsilon$.
We prove in Theorem 1 in Appendix$\S A.1$ that under the above assumptions, and the definitions above and those made in $\S A.1$, for

$$K \geq max \left\{ \frac{nL_1^4 C^4 (60)^2}{(1-q)^4}, \frac{nL_1^4 C^4 P_1}{(1-q)^2 \left( \frac{f\left(\frac{1}{n} \sum_l^n w_l^{(0)} x_l^{(0)}\right) - f^*}{\sqrt{nK}} \right)^2}, \frac{nP_2}{\frac{f\left(\frac{1}{n} \sum_l^n w_l^{(0)} x_l^{(0)}\right) - f^*}{\sqrt{nK}}} \right\} \quad (3)$$

The weighted mean of the local models converges in at least time $K$, that is,

$$\frac{1}{4} \frac{\sum_{k=0}^{K-1} \mathbb{E} \left\| \nabla f \left( \frac{1}{n} \sum_{l=1}^n w_l^{(k)} x_l^{(k)} \right) \right\|^2}{K} \leq 3 \frac{f\left(\frac{1}{n} \sum_l^n w_l^{(0)} x_l^{(0)}\right) - f^*}{\sqrt{nK}} \quad (4)$$

## 3 IMPLEMENTATION

We have performed our experiments on two graph topologies - $k$-regular and power-law graphs. Every node in the $k$-regular graph was made to communicate with $k - 1$ other neighbors in order to do a $k$-gossip averaging in every round. In order to make the *in_degree* (the number of nodes from which updates are received, including the self node) equal to the *out_degree* (the number of neighbors to which one's updates are sent), the nodes were simulated to be situated around a ring, and a symmetric communication was established between a node with $\lceil \frac{k-1}{2} \rceil$ nodes ahead of it and $\lfloor \frac{k-1}{2} \rfloor$ nodes behind it. In a power-law graph, some nodes are popular with many connections, while

Table 1: Training parameters

| Dataset | MNIST | FMNIST | Shakespeare |
|---|---|---|---|
| Model | DNN | DNN | LSTM |
| batch size | 32 | 32 | 32 |
| learning rate | 0.01 | 0.01 | 0.1 |
| n_rounds | 500 | 500 | 1000 |

a large number of nodes have only few connections. It is difficult to simulate a symmetric graph where both the *in_degree* and *out_degree* strictly follow the power law, hence an asymmetric graph was constructed with only the *in_degree* of nodes adhering strictly to the power-law in order to test the effectiveness of P2PRISM aggregation. To simulate this, the minimum degree was chosen as user input $k$ while the maximum *in_degree* ($deg_0$) equals $m/2$, where $m$ is the total number of nodes in the network. 1 random node would be simulated with *in_degree* = $deg_0$, 2 with $deg_1$, 4 with $deg_2$, and so on, where $k = deg_{log_2 m} < deg_{log_2 m - 1} < ... < deg_0 = m/2$ are in a geometric progression. There would be 1 another node with *in_degree* = $m$ to ensure that the *in_degree* distribution is in adherence with the power-law and all nodes have been assigned some *in_degree* value from the available degree values.

$c$ out of $m$ nodes were randomly chosen to be malicious with their models crafted with the SHE-JWALKAR attack with *unit vector* perturbation in every iteration. The source code was obtained from `https://github.com/vrt1shjwlkr/NDSS21-Model-Poisoning` and the same attack parameter ($\lambda_0 = 10.0$) was used. The gradient descent direction was estimated by accessing the current and past average of the models from the benign nodes. The attack assumes a mean-like aggregation being used to optimize its $\lambda$ value.

Every node was made to perform one iteration of local training ($l = 1$) on its minibatch, after which one round of gossip was performed. We use the image datasets of MNIST and Fashion-MNIST Xiao et al. (2017), and the NLP dataset of Shakespeare Caldas et al. (2019) for our experiments, trained with a DNN and an LSTM respectively. We report the test accuracy for the image dataset and the test perplexity ($= 2^{\text{test loss}}$) for the NLP dataset as used in the standard practice. The training hyperparameters are described in Table 1.

The DNN used to train MNIST and FMNIST consists of two CNN layers with 30 and 50 channels respectively, comprising of 3x3 kernels, the two layers being separated by a 2x2 maxpool layer, and followed by two fully connected layers of 200 and 10 neurons. The LSTM used to train Shakespeare consists of 1 hidden layer with 128 neurons between the input and the output layers. For the defense, a default value of $\mu = 0.75$ was used against the graybox attack and $\mu = 0$ against the whitebox attack. Due to the small value of $k$ used in simulations, we use the 0-percentile, that is the minimum flip-score value to set the upper bound of permissible flip-score. For comparison, the benign baseline of gossip averaging was used. In the malicious case, the baseline used was Trimmed Mean (TM) aggregation with $c_{max,i} = \lceil \frac{c*k_i}{m} \rceil$ for every node $i$ where $k_i$ is its number of neighbors. This trims the $c_{max,i}$ number of received gradient values from both the higher and lower end out of the $k_i$ values received based on their magnitude for every model weight and bias. The gradients here refer to the difference between the current received model from a neighbor and one's own model from the past iteration. By default, we choose $m = 128$ and the data was distributed in a non-IID manner with a bias $b$ of 0.5 for the image datasets, both of which had 10 classes. The nodes were divided into 10 groups and a data sample with label $l$ was assigned to group index $l$ with a probability $b$ and to any other group with a probability $\frac{1-b}{9}$. Within the group, the data samples were distributed randomly among the clients. For the NLP dataset, the data was divided sequentially (which is also non-IID) into $m + 1$ chunks, one for each client and the last chunk for testing.

## 4 EVALUATION

We first demonstrate how P2PRISM stops the spread of a graybox attack - one that only knows that a mean-like aggregation is being used but is oblivious to the details of the defense; and helps the benign nodes recover. Then we show how P2PRISM keeps the consensus distance of the graph under control which is most necessary for decentralized peer-to-peer learning. We then proceed to stress test P2PRISM under a whitebox attack where the attacker also has access to the defense

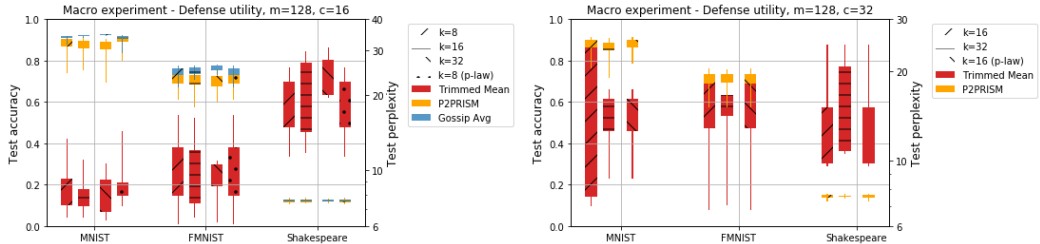

Figure 2: The figures show the final test accuracy (perplexity for Shakespeare) candles of the benign nodes comparing Trimmed Mean and P2PRISM under malicious conditions - $c/m = 1/8$ in the left and $c/m = 1/4$ in the right figure, along with the baseline benign results for Gossip Averaging in the left figure. For every dataset, the last bar corresponds to a power-law graph while the rest denote k-regular graphs.

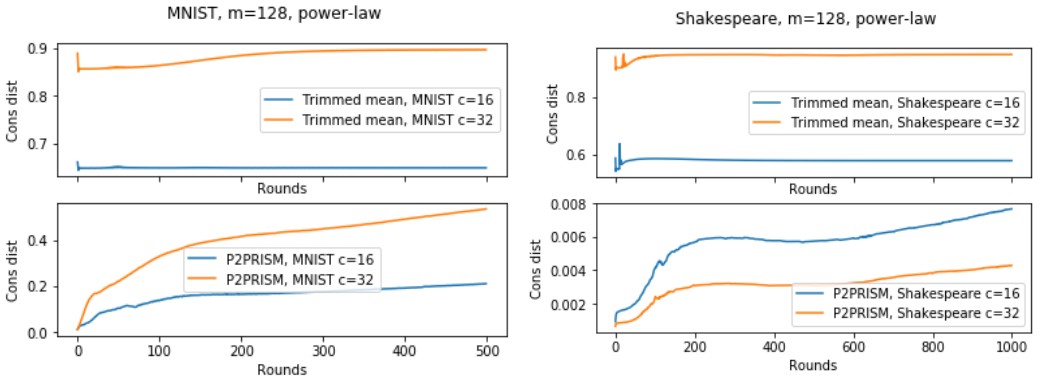

Figure 3: The figures show how consensus distance among the benign nodes changes during training. We show the results on one image dataset - MNIST on the left and on an NLP dataset - Shakespeare on the right. The graph topology chosen was power-law to mimic a realistic setting with $k = 8$ for $c = 16$ and $k = 16$ for $c = 32$. It is very clear how P2PRISM keeps the consensus control low as compared to Trimmed mean in a malicious environment.

algorithm, its parameters, and the cutoff flip-score in the past iteration beyond which all models were penalized. We find that the flip-score generated by malicious clients in the graybox attack are easily distinguishable from the benign ones due to the flip-score value being too large than the benign values and the detection rate is not affected by the defense parameter $\mu$ at all. Therefore, we use the whitebox attack to show the effect of varying $\mu$ where the malicious and benign flip-scores are indistinguishable.

Figure 2 shows the candle plot of test accuracy under different conditions. The lower and upper ends of the candle correspond to the 25-th and 75-th percentile final test accuracy (or test perplexity for Shakespeare dataset) respectively among the nodes, and the ends of the candle wicks correspond to the lowest and highest final test accuracy at the end of training. The attack completely damages P2PL without any secure aggregation to a random test accuracy of 10%. This is also evident from the level of impact that just 2 malicious nodes had in Figure 1. We do not show that in the plots here. We see that Trimmed Mean fails against the attack because of two reasons - 1) Magnitude based filtering is inefficient against a gradient ascent attack, and 2) a fixed value of $c_{max}$ although suitable for FL does not work for P2PL where the fraction of malicious nodes differs in every neighborhood in any practical case. We also observe a clear separation between the performance of Trimmed Mean and P2PRISM where P2PRISM is very close to the benign standard of Gossip Averaging, and is in some cases even completely overlapping with it. A small dip in the test accuracy could be explained from the fact that the malicious nodes attacked in every iteration and hence nothing was learned from their local data which did contribute in the completely benign case. It is important to observe that P2PRISM excels for both the graph types - k-regular and power-law across all the 3 datasets used.

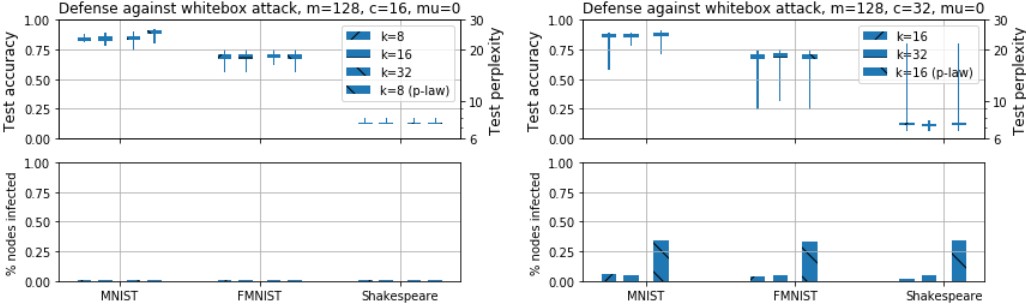

Figure 4: The figures show the performance of P2PRISM against a whitebox attacker that crafts a stealthy attack with full knowledge of the defense and its parameters to bypass the prism used in the defense. We consider the attack successful if it drops the test accuracy on MNIST below 0.8, and on FMNIST below 0.65, or if it raises the test perplexity on Shakespeare above 8. We observe that with $c = 16$, P2PRISM was successful in preventing all benign nodes from being infected. With $c = 32$, P2PRISM could restrict the number of infected nodes below 6% for a k-regular graph and under 34% for a power-law graph.

Next, we show how P2PRISM keeps the consensus among the benign nodes under control even in the presence of the attackers. In P2PL, every node can have a different model state, but as long as they are close to each other, they are all moving along the same trajectory towards the same global solution. We set the nodes to be at consensus initially. Naturally, as they train, they are going to drift from each other unless the graph is fully connected. The attackers try to spread dissensus among the benign nodes by infecting their neighbors. After a certain point, even if the attack is stopped, the existing dissensus may inhibit training as the nodes with disagreeing models collaborate. P2PRISM shows how, as in Figure 3, by identifying and filtering the malicious models based on one's flip-score, such a dissensus scenario is prevented. Having shown the success of P2PRISM against a graybox attack where the attacker was not aware of the details pf the defense being used by the nodes, we present a whitebox attacker at time $t$ that not only has access to the defense parameter $\mu$, but also the minimum, median, and cutoff flip-score of all the clients at $t - 1$. The attacker uses this information to craft the attack at time $t$ by optimizing the attack parameter $\lambda$ so that the expected flip-score at $t$ will be less than the cutoff flip-score at time $t - 1$ for the node being targeted. This is a highly personalized attack and the information from the past, that is, from $t - 1$ is used because the median, minimum, and cutoff values at $t$ is determined by a node only after receiving all the updates, while the attacker is yet to craft its attack. Given that every node just runs one iteration of local training, and the training trajectory is smooth, the estimation done by the attacker based on one timestep in the past is extremely accurate. The malicious and benign flip-scores are now indistinguishable and the defense is bypassed. However, we observe that this stealthy attack loses its damaging impact as a trade-off most of the time. We see in Figure 4 that P2PRISM could save 100% of the nodes from being infected when $c/m = 16/128 = 1/8$. However, on increasing the number of attackers to 32, that position themselves at random locations in the graph, the attackers do form a majority at some neighborhoods and are able to attack the benign nodes at such vulnerable locations. This number is very small though, as can be seen in Figure 4, that the attacker could infect less than 6% of the benign nodes in a k-regular graph and less than 34% in a power-law graph. Although the power-law graph seems to be infected, it is only because the attack, being whitebox, stealthy, and controlling 1/4-th of the popular nodes becomes too powerful. This shows the limit of malice upto which P2PRISM can be expected to provide perfect security, after which, saving all nodes from being infected cannot be guaranteed. $\mu$ was set to zero to allow the defense to be conservative against an attacker which is too powerful.

We take this opportunity to also demonstrate the effect of sweeping $\mu$. With higher $\mu$, the flip-score cutoff increases, thereby giving the attackers an opportunity to be stealthy even with higher attack magnitude $\lambda$. Thus, the stealthy attack may still be damaging as the detection threshold is not as sensitive anymore. We see the same trend in Figure 5 where the whitebox attackers are able to impact more and more nodes as $\mu$ increases, however this increase is too gradual and it reflects the robustness of P2PRISM. This also empirically shows why we have chosen a default value of $\mu = 0.75$ for all our experiments with the graybox attacker. It is always recommended to use a low

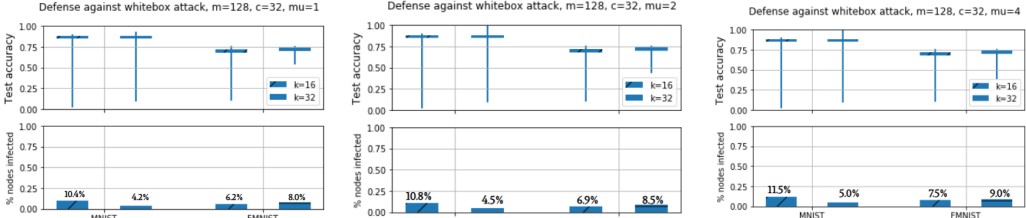

Figure 5: The figures show the effect of sweeping the defense parameter $\mu$ against a whitebox attack on MNIST and FMNIST in a k-regular graph with $m = 128, c = 32$. With increasing $\mu$, the cutoff flip-score increases for every node and a relatively stronger attack can get the opportunity to be stealthy and bypass P2PRISM. Hence, we see a gradual increase in percentage of infected nodes. However, even with a high value of $mu = 4$, this increase is very slow the fraction of infected nodes stay below 12%, thus proving the robustness of P2PRISM, that is, low sensitivity with respect to this parameter across a large sweep of its parameter.

$\mu$ for a conservative defense, however, this experiment proves that the performance of P2PRISM is not too sensitive with respect to its parameter $\mu$.

## 5   CONCLUSION

In this paper, we have presented a secure aggregation for nodes participating in collaborative learning as each others' peers. We began by introducing the concept of peer-to-peer learning in comparison with federated learning. We saw how the consensus distance at model initialization can significantly affect the training process and stressed on initializing all the participating nodes with the same model. This is an extremely important aspect of the training process but has been ignored in the context of decentralized collaborative learning. We also discussed how P2PL is differently affected by byzantine nodes as compared to FL depending on the graph connectivity, the number of gossiping steps per round, and the location of the malicious nodes. Unlike FL, an attacker can have full access to its neighbors' models via gossiping and can launch a personalized attack. At the same time, every node has access to its own trusted benign gradient updates to compare all incoming updates with, and unlike FL, a node can perform a trusted secure aggregation for itself, given a robust defense policy. We leveraged this fact to construct our own defense policy, P2PRISM based on the concept of flip-score that can protect a node even if its surrounded by a malicious majority in its neighborhood. We evaluated P2PRISM against the state-of-the-art model poisoning attack and compared it against the secure aggregation protocol of Trimmed Mean by extending it to the decentralized learning domain. P2PRISM could successfully recover all benign models under a graybox attacker, while also maintaining a low consensus distance among the benign nodes across all the three datasets of MNIST, FMNIST, and Shakespeare on both the k-regular and power-law graph topologies. This was followed by a stress test of P2PRISM against a whitebox attacker that also has access to the defense parameter in addition to access to all the benign models, that can launch a highly personalized attack to each of its neighbors. We saw that P2PRISM could recover all the being node even in the presence of this extremely powerful attacker when only 1/8-th of the nodes were malicious. However, when the fraction of malicious nodes go up to 1/4, some benign nodes begin to be infected, but their fraction is contained to less than 6% for a k-regular graph. With a power-law graph in the presence of such a powerful adversary controlling 1/4-th of all the nodes, P2PRISM could only save 66% of the nodes from being infected. We also swept the defense parameter under the presence of this whitebox attacker to show the robustness of P2PRISM across a large usable range of the parameter. P2PRISM could successfully contain the infection within 12% of the benign nodes. With this, we conclude the evaluation of P2PRISM and propose its usage at every node in a decentralized learning architecture for the best possible security against model poisoning attacks.

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

## A   APPENDIX

### A.1   CONVERGENCE ANALYSIS

**Theorem 1:**   Here, we show that, under P2PRISM, the weighted mean of the local models converges in at least time $K$, that is,

$$\frac{1}{4} \frac{\sum_{k=0}^{K-1} \mathbb{E} \left\| \nabla f \left( \frac{1}{n} \sum_{l=1}^{n} w_l^{(k)} x_l^{(k)} \right) \right\|^2}{K} \leq 3 \frac{f \left( \frac{1}{n} \sum_l^n w_l^{(0)} x_l^{(0)} \right) - f^*}{\sqrt{nK}} \tag{5}$$

where,

$$K \geq max \left\{ \frac{n L_1^4 C^4 (60)^2}{(1-q)^4}, \frac{n L_1^4 C^4 P_1}{(1-q)^2 \left( \frac{f \left( \frac{1}{n} \sum_l^n w_l^{(0)} x_l^{(0)} \right) - f^*}{\sqrt{nK}} \right)^2}, \frac{n P_2}{\frac{f \left( \frac{1}{n} \sum_l^n w_l^{(0)} x_l^{(0)} \right) - f^*}{\sqrt{nK}}} \right\} \tag{6}$$

For Assumptions 1-3 to be valid, for the sake of analysis, we assume all $n$ nodes to act benign within bounded variance, and we show that the standard decentralized algorithm, $\overline{x}^{(k)} = \frac{1}{n} \sum_{i=1}^{n} x_i^{(k)}$ achieves convergence under mixing matrices generated by P2PRISM. We modify the convergence analysis in Assran et al. (2019) by replacing a column stochastic mixing matrix with a row stochastic one, under the protocols proposed in P2PRISM. We describe the complete proof in great detail below.

**Theorem 2:**   Under the same assumptions in Theorem 1,

$$\frac{1}{nK} \sum_{k=0}^{K-1} \sum_{i=1}^{n} \mathrm{E} \left\| \frac{1}{n} \sum_{l}^{n} w_l^{(k)} x_l^{(k)} - \boldsymbol{x}_i^{(k)} \right\|^2 \leq \mathcal{O} \left( \frac{1}{K} + \frac{1}{K^{3/2}} \right),$$

$$\frac{1}{K} \sum_{k=0}^{K-1} \frac{1}{n} \sum_{i=1}^{n} \mathbb{E} \left\| \nabla f \left( \boldsymbol{x}_i^k \right) \right\|^2 \leq \mathcal{O} \left( \frac{1}{\sqrt{nK}} + \frac{1}{K} + \frac{1}{K^{3/2}} \right) \tag{7}$$

That is, the parameters at each node converge as well.

**Lemma 3:**   *Under Assumptions 1-3, let* $\lambda = 1 - nD^{-\Delta B}$ *and* $q = \lambda^{\frac{1}{\Delta B + 1}}$, *then there exists a constant* $C < \frac{2\sqrt{d} D^{\Delta B}}{\lambda^{\frac{\Delta B + 2}{\Delta B + 1}}} \left( 1 + \frac{n^{3/2}}{2} \right)$ *where* $d$ *is the dimension of* $x_i^{(k)}$ *for all* $i = 1, 2, ..., n$ *and* $k \geq 0$. $n$ *is the number of nodes. The communication topology is* $B$-*strongly connected. To each mixing matrix* $\mathbf{P}^{(k)}$, *we can associate a graph with vertex set* $\{1, \ldots, n\}$ *and edge set* $E^{(k)} = \{(i, j) : w_{i.j}^{(k)} > 0\}$. $B$-*strongly connected means that we can assume that there exists finite, positive integers,* $B$ *and* $\Delta$, *such that the graph with edge set* $\bigcup_{k=lB}^{(l+1)B-1} E^{(k)}$ *is strongly connected and has diameter at most* $\Delta$ *for every* $l \geq 0$. *Then*

$$\left\| \frac{1}{n} \sum_{l=1}^{n} w_l^{(k)} x_l^{(k)} - x_i^{(k)} \right\|_2 \leq C q^k \left\| x_i^{(0)} \right\|_2 + \gamma C \sum_{s=0}^{k} q^{k-s} \left\| \nabla F_i \left( x_i^{(s)}; \xi_i^{(s)} \right) \right\|_2 \tag{8}$$

***Proof:***   We make modifications in Lemma 3 in Assran et al. (2019) to prove our inequality. It is to be noted that former analysis assumes column stochastic mixing matrices, which have the property that when premultiplied by a row of ones ($\mathbf{1}_n^T$), the product equals $\mathbf{1}_n^T$. Whereas, in our case, such an operation results in a vector $w$ each element of which represents the column sum of the mixing matrix. We prove in the following lemmas how this modification affects the convergence analysis. Lemma 3 in Assran et al. (2019) says -
Suppose that Assumption 3 (mixing connectivity) holds. Let $\lambda = 1 - nD^{-(\tau+1)\Delta B}$ and let $q = \lambda^{1/((\tau+1)\Delta B + 1)}$ where $\tau$ is the delay in asynchronous communication among the nodes. Then there exists a constant $C < \frac{2\sqrt{d} D^{(\tau+1)\Delta B}}{\lambda^{\frac{(\tau+1)\Delta B + 2}{(\tau+1)\Delta B + 1}}}$, where $d$ is the dimension of $\overline{x}^{(k)}$, $z_i^{(k)}$, and $x_i^{(0)}$, such that, for

all $i = 1, 2, \ldots, n$ (non-virtual nodes) and $k \geq 0$ such that

$$\left\| \overline{x}^{(k)} - z_i^{(k)} \right\|_2 \leq Cq^k \left\| x_i^{(0)} \right\|_2 + \gamma C \sum_{s=0}^{k} q^{k-s} \left\| \nabla F_i \left( z_i^{(s)}; \xi_i^{(s)} \right) \right\|_2 \tag{9}$$

where $z$ is the unbiased estimate of $x$ in their problem formulation. This lemma itself follows from a small adaptation to Theorem 1 in Assran & Rabbat (2020) that is proven in Assran (2018). We show the modifications we make in the former proof to achieve our result. It is shown in Assran (2018) that

$$\left\| z_i^{(k+1)} - \frac{\mathbf{1}^\top x^{(k)}}{n} \right\|_1 \leq \frac{C}{\delta_{\min}} q^k \| x^{(0)} \|_1 + \frac{C}{\delta_{\min}} \left( \sum_{s=1}^{k} q^{k-s} \| \eta^{(s)} \|_1 + \left\| \frac{\mathbf{1}^\top x^{(k)}}{n} \right\|_1 q^k \right) \tag{10}$$

where $\eta$ is the local update made by a node after gossip averaging as shown in the expression below.

$$
\begin{aligned}
x^{(1)} &= P^{(0)} x^{(0)} + \eta^{(1)} \\
x^{(2)} &= P^{(1)} x^{(1)} + \eta^{(2)} \\
&= P^{(1)} P^{(0)} x^{(0)} + P^{(1)} \eta^{(1)} + \eta^{(2)}
\end{aligned}
$$

$$\text{In general, } x^{(k)} = \Pi_{i=0}^{k-1} P^{(i)} x^{(0)} + \sum_{i=1}^{k-1} \Pi_{j=i}^{k-1} P^{(j)} \eta^{(i)} + \eta^{(k)} \tag{11}$$

If $P$ is column stochastic, then $\mathbf{1}^T P = \mathbf{1}^T$. This fact is used to simplify the above equation by premultiplying LHS and RHS by $\mathbf{1}^T$ to obtain $\mathbf{1}^T x^{(k)} = \mathbf{1}^T x^{(0)} + \sum_{i=1}^{k} \mathbf{1}^T \eta^{(i)}$, which leads to the inequality $\left\| \mathbf{1}^T x^{(k)} \right\| \leq \left\| \mathbf{1}^T x^{(0)} \right\| + \sum_{s=1}^{k} \left\| \mathbf{1}^T \eta^{(s)} \right\| \leq n \left\| x^{(0)} \right\| + n \sum_{s=1}^{k} \left\| \eta^{(s)} \right\|$. This when plugged in 10 leads to 9 with $C = \frac{2C}{\delta_{min}}$. We modify the above analysis by replacing $\mathbf{1}^T$ by $w^{(k)}$ where $\sum_{i=1}^{n} w_i^{(k)} = n$. Premultiplying 11 by $w^{(k)}$ results in the equation

$$w^{(k)} x^{(k)} = w^{(k)} \Pi_{i=0}^{k-1} P^{(i)} x^{(0)} + w^{(k)} \sum_{i=1}^{(k-1)} \Pi_{j=i}^{k-1} P^{(j)} \eta^{(i)} + w^{(k)} \eta^{(k)}$$

We now use the fact that *the product of two row-stochastic matrices is also row-stochastic*, and we represent all products of matrices above by a single matrix, simplified as

$$w^{(k)} x^{(k)} = w^{(k)} P_0 x^{(0)} + w^{(k)} \sum_{i=1}^{k-1} P_i \eta^{(i)} + w^{(k)} \eta^{(k)}$$

Absorbing the rightmost term into the summation assuming an identity matrix P (that is also row stochastic) and applying the triangle inequality, we obtain -

$$\left\| w^{(k)} x^{(k)} \right\| \leq \left\| w^{(k)} P_0 x^{(0)} \right\| + \sum_{i=1}^{k} \left\| w^{(k)} P_i \eta^{(i)} \right\|$$

**Upper bound on elements of $P$**   In our construction, the mixing matrix $P$ is row stochastic, and every node gives equal weights to all its neighbors that pass its $PRISM$. Assuming every node chooses to aggregate the model from at least one of its neighbors in every iteration, allotting a weight of $\frac{1}{2}$ to itself and the neighbor, therefore the upper limit on the column sum $w_i$ is $\frac{n}{2}$ for a node $i$. This happens when every node $j$ chooses to aggregate only two models, that are, $x_i$ and $x_j$, and allot of weight of 0 to all other nodes. Thus, if we consider the vector $w' = wP$, where $P$ is any row-stochastic matrix and $w$ is the column sum of any row-stochastic matrix, then the $i - th$ element $w_i'$ equals $\sum_j w_j c_{i,j}$ where $c_i$ is the $i - th$ column of $P$. We have the $j - th$ element of $w'$, $\sum_j w_j c_{i,j} \leq \left( \sum_j w_j \right) \left( \sum_j c_{i,j} \right) = n \left( \frac{n}{2} \right)$. Hence, for all such $w, P$, we have $\| wP \|^2 \leq n \left( \frac{n}{2} \right)^2$, that is, $\frac{\| wP \|}{n} \leq \frac{n^{5/2}}{2}$.

We obtain that $\frac{\| w^{(k)} x^{(k)} \|}{n} \leq \frac{n^{3/2}}{2} \left\| x^{(0)} \right\| + \frac{n^{3/2}}{2} \sum_{s=1}^{k} \left\| \eta^{(s)} \right\|$. Plugging this in 10, we obtain back 9 with $C = \frac{C}{\delta_{min}} \left( 1 + \frac{n^{3/2}}{2} \right)$. Therefore, the lemma described in Assran et al. (2019) for column

stochastic matrices remains valid for a row-stochastic mixing matrix as well but for a different constant $C$. Now we substitute $\tau = 0$ since our setup is synchronous to present the upper bound on our constant C, that is, $C < \frac{2\sqrt{d}D^{\Delta B}}{\lambda^{\frac{\Delta B+2}{\Delta B+1}}} \left(1 + \frac{n^{3/2}}{2}\right)$, where $\lambda = 1 - nD^{-\Delta B}$ and $q = \lambda^{\frac{1}{\Delta B+1}}$. Since in our setup. $z_i = x_i$, we substitute it back to complete our proof.

**Lemma 4:** *(Bound of stochastic gradient). We have the following inequality under Assumptions 1 and 2:*

$$\mathbb{E}\left\|\nabla f_i\left(x_i^{(k)}\right)\right\|^2 \leq 3L^2 \mathbb{E}\left\|x_i^{(k)} - \frac{1}{n}\sum_{l=1}^{n} w_l^{(k)} x_l^{(k)}\right\|^2 + 3\zeta^2 + 3\mathbb{E}\left\|\nabla f\left(\frac{1}{n}\sum_{l=1}^{n} w_l^{(k)} x_l^{(k)}\right)\right\|^2 \quad (12)$$

*Proof:*

$$\mathbb{E}\left\|\nabla f_i\left(x_i^{(k)}\right)\right\|^2 \overset{\text{Cauchy-Schwartz}}{\leq} 3\mathbb{E}\left\|\nabla f_i\left(x_i^{(k)}\right) - \nabla f_i\left(\frac{1}{n}\sum_{l=1}^{n} w_l^{(k)} x_l^{(k)}\right)\right\|^2$$

$$+ 3\mathbb{E}\left\|\nabla f_i\left(\frac{1}{n}\sum_{l=1}^{n} w_l^{(k)} x_l^{(k)}\right) - \nabla f\left(\frac{1}{n}\sum_{l=1}^{n} w_l^{(k)} x_l^{(k)}\right)\right\|^2 + 3\mathbb{E}\left\|\nabla f\left(\frac{1}{n}\sum_{l=1}^{n} w_l^{(k)} x_l^{(k)}\right)\right\|^2$$

$$\overset{\text{L-smooth}}{\leq} 3L^2 \mathbb{E}\left\|x_i^{(k)} - \frac{1}{n}\sum_{l=1}^{n} w_l^{(k)} x_l^{(k)}\right\|^2 + 3\mathbb{E}\left\|\nabla f_i\left(\frac{1}{n}\sum_{l=1}^{n} w_l^{(k)} x_l^{(k)}\right) - \nabla f\left(\frac{1}{n}\sum_{l=1}^{n} w_l^{(k)} x_l^{(k)}\right)\right\|^2$$

$$+ 3\mathbb{E}\left\|\nabla f\left(\frac{1}{n}\sum_{l=1}^{n} w_l^{(k)} x_l^{(k)}\right)\right\|^2$$

$$\overset{\text{Bounded Variance}}{\leq} 3L^2 \mathbb{E}\left\|x_i^{(k)} - \frac{1}{n}\sum_{l=1}^{n} w_l^{(k)} x_l^{(k)}\right\|^2 + 3\zeta^2 + 3\mathbb{E}\left\|\nabla f\left(\frac{1}{n}\sum_{l=1}^{n} w_l^{(k)} x_l^{(k)}\right)\right\|^2$$

$$(13)$$

**Lemma 5:** *Under Assumptions 1-3, we have*

$$Q_i^{(k)} = \mathbb{E}\left\|\frac{1}{n}\sum_{l}^{n} w_l^{(k)} x_l^{(k)} - x_i^{(k)}\right\|^2 \leq \left(\gamma^2 \frac{4C^2}{(1-q)^2} + \gamma\frac{q^k C^2}{1-q}\right)\sigma^2 + \left(\gamma^2 \frac{12C^2}{(1-q)^2} + \gamma\frac{q^k 3C^2}{1-q}\right)\zeta^2$$

$$+ \left(\gamma^2 \frac{12L^2 C^2}{1-q} + \gamma q^k 3L^2 C^2\right)\sum_{j=0}^{k} q^{k-j} Q_i^{(j)}$$

$$+ \left(\gamma^2 \frac{12C^2}{1-q} + \gamma q^k 3C^2\right)\sum_{j=0}^{k} q^{k-j} \mathbb{E}\left\|\nabla f\left(\sum_{l}^{n} w_l^{(j)} x_l^{(j)}\right)\right\|^2$$

$$+ \left(q^{2k} C^2 + \gamma q^k \frac{2C^2}{1-q}\right)\left\|x_i^{(0)}\right\|^2.$$

$$(14)$$

***Proof:*** Let us define,

$$
\begin{aligned}
Q_i^{(k)} =& \mathbb{E}\left\|\frac{1}{n}\sum_l^n w_l^{(k)} x_l^{(k)} - x_i^{(k)}\right\|^2 \\
\overset{\text{Lemma 3}}{\leq}& \mathbb{E}\left(Cq^k\left\|x_i^{(0)}\right\| + \gamma C \sum_{s=0}^k q^{k-s}\left\|\nabla F_i\left(x_i^{(s)};\xi_i^{(s)}\right)\right\|\right)^2 \\
=& \mathbb{E}\left(Cq^k\left\|x_i^{(0)}\right\| + \gamma C \sum_{s=0}^k q^{k-s}\left\|\nabla F_i\left(x_i^{(s)};\xi_i^{(s)}\right) - \nabla f_i\left(x_i^{(s)}\right) + \nabla f_i\left(x_i^{(s)}\right)\right\|\right)^2 \\
\leq& \mathbb{E}(\underbrace{Cq^k\left\|x_i^{(0)}\right\|}_{a} + \underbrace{\gamma C \sum_{s=0}^k q^{k-s}\left\|\nabla F_i\left(x_i^{(s)};\xi_i^{(s)}\right) - \nabla f_i\left(x^{(s)}\right)\right\|}_{b} + \underbrace{\gamma C \sum_{s=0}^k q^{k-s}\left\|\nabla f_i\left(x_i^{(s)}\right)\right\|}_{c})^2
\end{aligned}
$$

$$(15)$$

Thus, using the above expressions of $a$, $b$ and $c$ we have that $Q_i^{(k)} \leq \mathbb{E}\left(a^2 + b^2 + c^2 + 2ab + 2bc + 2ac\right)$. Let us now obtain bounds for all of these quantities:

$$
\begin{aligned}
a^2 =& C^2\left\|x_i^{(0)}\right\|^2 q^{2k} \\
b^2 =& \gamma^2 C^2 \sum_{j=0}^k q^{2(k-j)}\left\|\nabla F_i\left(x_i^{(j)};\xi_i^{(j)}\right) - \nabla f_i\left(x_i^{(j)}\right)\right\|^2 \\
& + \underbrace{2\gamma^2 C^2 \sum_{j=0}^k \sum_{s=j+1}^k q^{2k-j-s}\left\|\nabla F_i\left(x_i^{(j)};\xi_i^{(j)}\right) - \nabla f_i\left(x_i^{(j)}\right)\right\|\left\|\nabla F_i\left(x_i^{(s)};\xi_i^{(s)}\right) - \nabla f_i\left(x_i^{(s)}\right)\right\|}_{c^2} \\
c^2 =& \gamma^2 C^2 \sum_{j=0}^k q^{2(k-j)}\left\|\nabla f_i\left(x_i^{(j)}\right)\right\|^2 + \underbrace{2\gamma^2 C^2 \sum_{j=0}^k \sum_{s=j+1}^k q^{2k-j-s}\left\|\nabla f_i\left(x_i^{(j)}\right)\right\|\left\|\nabla f_i\left(x_i^{(s)}\right)\right\|}_{c_1} \\
2ab =& 2\gamma C^2 q^k\left\|x_i^{(0)}\right\| \sum_{s=0}^k q^{k-s}\left\|\nabla F_i\left(x_i^{(s)};\xi_i^{(s)}\right) - \nabla f_i\left(x_i^{(s)}\right)\right\| \\
2ac =& 2\gamma C^2 q^k\left\|x_i^{(0)}\right\| \sum_{s=0}^k q^{k-s}\left\|\nabla f_i\left(x_i^{(s)}\right)\right\| \\
2bc =& 2\gamma^2 C^2 \sum_{j=0}^k \sum_{s=0}^k q^{2k-j-s}\left\|\nabla F_i\left(x_i^{(j)};\xi_i^{(j)}\right) - \nabla f_i\left(x_i^{(j)}\right)\right\|\left\|\nabla f_i\left(x_i^{(s)}\right)\right\|
\end{aligned}
$$

The expression $b_1$ is bounded as follows:

$$
\begin{aligned}
b_1 &= \gamma^2 C^2 \sum_{j=0}^{k} \sum_{s=j+1}^{k} q^{2k-j-s} 2 \left\| \nabla F_i\left(x_i^{(j)}; \xi_i^{(j)}\right) - \nabla f_i\left(x_i^{(j)}\right) \right\| \left\| \nabla F_i\left(x_i^{(s)}; \xi_i^{(s)}\right) - \nabla f_i\left(x_i^{(s)}\right) \right\| \\
&\leq \gamma^2 C^2 \sum_{j=0}^{k} \sum_{s=j+1}^{k} q^{2k-s-j} \left\| \nabla F_i\left(x_i^{(j)}; \xi_i^{(j)}\right) - \nabla f_i\left(x_i^{(j)}\right) \right\|^2 \quad \text{using } 2ab \leq a^2 + b^2. \\
&+ \gamma^2 C^2 \sum_{j=0}^{k} \sum_{s=j+1}^{k} q^{2k-s-j} \left\| \nabla F_i\left(x_i^{(s)}; \xi_i^{(s)}\right) - \nabla f_i\left(x_i^{(s)}\right) \right\|^2 \\
&\leq \gamma^2 C^2 \sum_{j=0}^{k} \sum_{s=0}^{k} q^{2k-s-j} \left\| \nabla F_i\left(x_i^{(j)}; \xi_i^{(j)}\right) - \nabla f_i\left(x_i^{(j)}\right) \right\|^2 \\
&+ \gamma^2 C^2 \sum_{j=0}^{k} \sum_{s=0}^{k} q^{2k-s-j} \left\| \nabla F_i\left(x_i^{(s)}; \xi_i^{(s)}\right) - \nabla f_i\left(x_i^{(s)}\right) \right\|^2 \\
&= \gamma^2 C^2 \sum_{j=0}^{k} q^{k-j} \left\| \nabla F_i\left(x_i^{(j)}; \xi_i^{(j)}\right) - \nabla f_i\left(x_i^{(j)}\right) \right\|^2 \sum_{s=0}^{k} q^{k-s} \\
&+ \gamma^2 C^2 \sum_{s=0}^{k} q^{k-s} \left\| \nabla F_i\left(x_i^{(s)}; \xi_i^{(s)}\right) - \nabla f_i\left(x_i^{(s)}\right) \right\|^2 \sum_{j=0}^{k} q^{k-j} \\
&\leq \frac{1}{1-q} \gamma^2 C^2 \sum_{j=0}^{k} q^{k-j} \left\| \nabla F_i\left(x_i^{(j)}; \xi_i^{(j)}\right) - \nabla f_i\left(x_i^{(j)}\right) \right\|^2 \quad \text{using } \sum_{k=0}^{K} r^K \leq \frac{1}{1-r} \text{ for } r < 1 \\
&+ \frac{1}{1-q} \gamma^2 C^2 \sum_{s=0}^{k} q^{k-s} \left\| \nabla F_i\left(x_i^{(s)}; \xi_i^{(s)}\right) - \nabla f_i\left(x_i^{(s)}\right) \right\|^2 \\
&= \frac{2}{1-q} \gamma^2 C^2 \sum_{j=0}^{k} q^{k-j} \left\| \nabla F_i\left(x_i^{(j)}; \xi_i^{(j)}\right) - \nabla f_i\left(x_i^{(j)}\right) \right\|^2.
\end{aligned}
$$

$$(16)$$

Thus,

$$
\begin{aligned}
b^2 &= \gamma^2 C^2 \sum_{j=0}^{k} q^{2(k-j)} \left\| \nabla F_i\left(x_i^{(j)}; \xi_i^{(j)}\right) - \nabla f_i\left(x_i^{(j)}\right) \right\|^2 + b_1 \\
&\leq \frac{\gamma^2 C^2}{1-q} \sum_{j=0}^{k} q^{k-j} \left\| \nabla F_i\left(x_i^{(j)}; \xi_i^{(j)}\right) - \nabla f_i\left(x_i^{(j)}\right) \right\|^2 + b_1 \\
&\overset{(16)}{\leq} \frac{3\gamma^2 C^2}{1-q} \sum_{j=0}^{k} q^{k-j} \left\| \nabla F_i\left(x_i^{(j)}; \xi_i^{(j)}\right) - \nabla f_i\left(x_i^{(j)}\right) \right\|^2
\end{aligned}
$$

$$(17)$$

By identical construction we have

$$
c^2 \leq \frac{3\gamma^2 C^2}{1-q} \sum_{j=0}^{k} q^{k-j} \left\| \nabla f_i\left(x_i^{(j)}\right) \right\|^2
$$

Now let us bound the products $2ab$, $2ac$ and $2bc$.

$$
\begin{aligned}
2ab =& \gamma C^2 q^k \sum_{s=0}^{k} q^{k-s} 2 \left\| x_i^{(0)} \right\| \left\| \nabla F_i \left( x_i^{(s)}; \xi_i^{(s)} \right) - \nabla f_i \left( x_i^{(s)} \right) \right\| \\
\leq& \gamma C^2 q^k \sum_{j=0}^{k} q^{k-j} \left\| \nabla F_i \left( x_i^{(j)}; \xi_i^{(j)} \right) - \nabla f_i \left( x_i^{(j)} \right) \right\|^2 + \gamma C^2 q^k \sum_{j=0}^{k} q^{k-j} \left\| x_i^{(0)} \right\|^2 \text{ using } 2ab \leq a^2 + b^2. \\
\leq& \gamma C^2 q^k \sum_{j=0}^{k} q^{k-j} \left\| \nabla F_i \left( x_i^{(j)}; \xi_i^{(j)} \right) - \nabla f_i \left( x_i^{(j)} \right) \right\|^2 + \frac{\gamma C^2 \left\| x_i^{(0)} \right\|^2}{1-q} q^k \text{ using } \sum_{k=0}^{K} r^K \leq \frac{1}{1-r} \text{ for } r < 1
\end{aligned}
\tag{18}
$$

By similar procedure,

$$
2ac \leq \gamma C^2 q^k \sum_{s=0}^{k} q^{k-s} \left\| \nabla f_i \left( x_i^{(s)} \right) \right\|^2 + \frac{\gamma C^2 \left\| x_i^{(0)} \right\|^2}{1-q} q^k
\tag{19}
$$

Finally,

$$
\begin{aligned}
2bc =& \gamma^2 C^2 \sum_{j=0}^{k} \sum_{s=0}^{k} q^{2k-j-s} 2 \left\| \nabla F_i \left( x_i^{(j)}; \xi_i^{(j)} \right) - \nabla f_i \left( x_i^{(j)} \right) \right\| \left\| \nabla f_i \left( x_i^{(s)} \right) \right\| \\
\leq& \gamma^2 C^2 \sum_{j=0}^{k} \sum_{s=0}^{k} q^{2k-j-s} \left\| \nabla F_i \left( x_i^{(j)}; \xi_i^{(j)} \right) - \nabla f_i \left( x_i^{(j)} \right) \right\|^2 + \gamma^2 C^2 \sum_{j=0}^{k} \sum_{s=0}^{k} q^{2k-j-s} \left\| \nabla f_i \left( x_i^{(s)} \right) \right\|^2, \\
=& \gamma^2 C^2 \sum_{j=0}^{k} q^{k-j} \left\| \nabla F_i \left( x_i^{(j)}; \xi_i^{(j)} \right) - \nabla f_i \left( x_i^{(j)} \right) \right\|^2 \sum_{s=0}^{k} q^{k-s} + \gamma^2 C^2 \sum_{s=0}^{k} q^{k-s} \left\| \nabla f_i \left( x_i^{(s)} \right) \right\|^2 \sum_{j=0}^{k} q^{k-j}, \\
\leq& \frac{\gamma^2 C^2}{1-q} \sum_{j=0}^{k} q^{k-j} \left\| \nabla F_i \left( x_i^{(j)}; \xi_i^{(j)} \right) - \nabla f_i \left( x_i^{(j)} \right) \right\|^2 + \frac{\gamma^2 C^2}{1-q} \sum_{s=0}^{k} q^{k-s} \left\| \nabla f_i \left( x_i^{(s)} \right) \right\|^2
\end{aligned}
\tag{20}
$$

By combining all of the above bounds together we obtain:

$$
\begin{aligned}
Q_i^{(k)} \leq& \mathbb{E} \left( a^2 + b^2 + c^2 + 2ab + 2bc + 2ac \right) \\
\leq& \mathbb{E} \frac{4\gamma^2 C^2}{1-q} \sum_{j=0}^{k} q^{k-j} \left\| \nabla F_i \left( x_i^{(j)}; \xi_i^{(j)} \right) - \nabla f_i \left( x_i^{(j)} \right) \right\|^2 \\
& + \mathbb{E} \frac{4\gamma^2 C^2}{1-q} \sum_{j=0}^{k} q^{k-j} \left\| \nabla f_i \left( x_i^{(j)} \right) \right\|^2 \\
& + C^2 \left\| x_i^{(0)} \right\|^2 q^{2k} \\
& + \frac{2\gamma C^2 \left\| x_i^{(0)} \right\|^2}{1-q} q^k \\
& + \mathbb{E} \gamma C^2 q^k \sum_{j=0}^{k} q^{k-j} \left\| \nabla f_i \left( x_i^{(j)} \right) \right\|^2 \\
& + \mathbb{E} \gamma C^2 q^k \sum_{j=0}^{k} q^{k-j} \left\| \nabla F_i \left( x_i^{(j)}; \xi_i^{(j)} \right) - \nabla f_i \left( x_i^{(j)} \right) \right\|^2
\end{aligned}
\tag{21}
$$

After grouping terms together and using the upper bound of Lemma 4, we obtain

$$
\begin{aligned}
Q_i^{(k)} \quad \leq \quad & \left( \gamma^2 \frac{4C^2}{(1-q)^2} + \gamma \frac{q^k C^2}{1-q} \right) \sigma^2 + \left( q^{2k} C^2 + \gamma q^k \frac{2C^2}{1-q} \right) \left\| x_i^{(0)} \right\|^2 . \\
+ \quad & \left( \gamma^2 \frac{4C^2}{1-q} + \gamma q^k C^2 \right) \sum_{j=0}^{k} q^{k-j} \mathbb{E} \left\| \nabla f_i \left( x_i^{(j)} \right) \right\|^2 \\
\overset{\text{Lemma 4}}{\leq} \quad & \left( \gamma^2 \frac{4C^2}{(1-q)^2} + \gamma \frac{q^k C^2}{1-q} \right) \sigma^2 + \left( q^{2k} C^2 + \gamma q^k \frac{2C^2}{1-q} \right) \left\| x_i^{(0)} \right\|^2 \\
+ \quad & \left( \gamma^2 \frac{12C^2}{(1-q)^2} + \frac{\gamma q^k 3C^2}{1-q} \right) \zeta^2 \\
+ \quad & \left( \gamma^2 \frac{12L^2 C^2}{1-q} + \gamma q^k 3L^2 C^2 \right) \sum_{j=0}^{k} q^{k-j} Q_i^{(j)} \\
+ \quad & \left( \gamma^2 \frac{12C^2}{1-q} + \gamma q^k 3C^2 \right) \sum_{j=0}^{k} q^{k-j} \mathbb{E} \left\| \nabla f \left( \frac{1}{n} \sum_{l}^{n} w_l^{(j)} x_l^{(j)} \right) \right\|^2
\end{aligned}
\tag{22}
$$

Having found a bound for the quantity $Q_i^{(k)}$, let us now present a lemma for bounding the quantity $\sum_{k=0}^{K-1} M^{(k)}$ where $K > 1$ is a constant and $M^{(k)}$ is the average $Q_i^{(k)}$ across all (non-virtual) nodes $i \in [n]$. That is, $M^{(k)} = \frac{1}{n} \sum_{i=1}^{n} Q_i^{(k)}$.

**Lemma 6:** *Let Assumptions 1-3 hold and let us define $D_2 = 1 - \gamma^2 \frac{12L^2 C^2}{(1-q)^2} - \gamma \frac{3L^2 C^2}{(1-q)^2}$. Then,*

$$
\begin{aligned}
\sum_{k=0}^{K-1} M^{(k)} \leq & \left( \gamma^2 \frac{4C^2}{(1-q)^2 D_2} \right) \sigma^2 K + \left( \gamma \frac{C^2}{(1-q)^2 D_2} \right) \sigma^2 \\
+ & \left( \gamma^2 \frac{12C^2}{(1-q)^2 D_2} \right) \zeta^2 K + \left( \frac{\gamma 3C^2}{(1-q)^2 D_2} \right) \zeta^2 \\
+ & \left( \frac{C^2}{(1-q)^2 D_2} + \gamma \frac{2C^2}{(1-q)^2 D_2} \right) \frac{\sum_{i=1}^{n} \left\| x_i^{(0)} \right\|^2}{n} \\
+ & \left( \gamma^2 \frac{12C^2}{(1-q)^2 D_2} + \gamma \frac{3C^2}{(1-q)^2 D_2} \right) \sum_{k=0}^{K-1} \mathbb{E} \left\| \nabla f \left( \frac{1}{n} \sum_{l}^{n} w_l^{(k)} x_l^{(k)} \right) \right\|^2
\end{aligned}
\tag{23}
$$

***Proof:*** Using the bound for $Q_i^{(k)}$, let us first bound its average across all nodes $M^{(k)}$.

$$
\begin{aligned}
M^{(k)} \quad = \quad & \frac{1}{n} \sum_{i=1}^{n} Q_i^{(k)} \\
\overset{\text{Lemma 5}}{\leq} \quad & \left( \gamma^2 \frac{4C^2}{(1-q)^2} + \gamma \frac{q^k C^2}{1-q} \right) \sigma^2 + \left( \gamma^2 \frac{12C^2}{(1-q)^2} + \frac{\gamma q^k 3C^2}{1-q} \right) \zeta^2 \\
+ \quad & \left( \gamma^2 \frac{12C^2}{1-q} + \gamma q^k 3C^2 \right) \sum_{j=0}^{k} q^{k-j} \mathbb{E} \left\| \nabla f \left( \frac{1}{n} \sum_{l}^{n} w_l^{(k)} x_l^{(k)} \right) \right\|^2 \\
+ \quad & \left( \gamma^2 \frac{12L^2 C^2}{1-q} + \gamma q^k 3L^2 C^2 \right) \sum_{j=0}^{k} q^{k-j} M^{(j)} \\
+ \quad & \left( q^{2k} C^2 + \gamma q^k \frac{2C^2}{1-q} \right) \frac{\sum_{i=1}^{n} \left\| x_i^{(0)} \right\|^2}{n} .
\end{aligned}
\tag{24}
$$

At this point note that for any $\lambda \in (0,1)$, non-negative integer $K \in \mathbb{N}$, and non-negative sequence $\left\{\beta^{(j)}\right\}_{j=0}^{k}$, it holds that

$$\sum_{k=0}^{K}\sum_{j=0}^{k} \lambda^{k-j}\beta^{(j)} = \beta^{(0)}\left(\lambda^K + \lambda^{K-1} + \cdots + \lambda^0\right) + \beta^{(1)}\left(\lambda^{K-1} + \lambda^{K-2} + \cdots + \lambda^0\right) + \cdots + \beta^{(K)}\left(\lambda^0\right)$$

$$\leq \frac{1}{1-\lambda}\sum_{j=0}^{K}\beta^{(j)}$$

(25)

Similarly,

$$\sum_{k=0}^{K}\lambda^k \sum_{j=0}^{k}\lambda^{k-j}\beta^{(j)} = \sum_{k=0}^{K}\sum_{j=0}^{k}\lambda^{2k-j}\beta^{(j)} \leq \sum_{k=0}^{K}\sum_{j=0}^{k}\lambda^{2(k-j)}\beta^{(j)} \overset{(18)}{\leq} \frac{1}{1-\lambda^2}\sum_{j=0}^{K}\beta^{(j)} \quad (26)$$

Now by summing from $k=0$ to $k = K-1$ and using the bounds of 25 and 26 we obtain

$$\begin{aligned}
\sum_{k=0}^{K-1} M^{(k)} \leq &\left(\gamma^2\frac{4C^2}{(1-q)^2}\right)\sigma^2 K + \left(\gamma\frac{C^2}{(1-q)^2}\right)\sigma^2 \\
&+ \left(\gamma^2\frac{12C^2}{(1-q)^2}\right)\zeta^2 K + \left(\frac{\gamma 3C^2}{1-q}\right)\zeta^2 \\
&+ \left(\frac{C^2}{1-q^2} + \gamma\frac{2C^2}{(1-q)^2}\right)\frac{\sum_{i=1}^{n}\left\|x_i^{(0)}\right\|^2}{n} \\
&+ \left(\gamma^2\frac{12C^2}{(1-q)^2} + \gamma\frac{3C^2}{1-q^2}\right)\sum_{k=0}^{K-1}\mathbb{E}\left\|\nabla f\left(\frac{1}{n}\sum_l^n w_l^{(k)}x_l^{(k)}\right)\right\|^2 \\
&+ \left(\gamma^2\frac{12L^2C^2}{(1-q)^2} + \gamma\frac{3L^2C^2}{1-q^2}\right)\sum_{k=0}^{K-1} M^{(k)}.
\end{aligned}$$

(27)

By rearranging:

$$\begin{aligned}
\left(1 - \gamma^2\frac{12L^2C^2}{(1-q)^2} - \gamma\frac{3L^2C^2}{1-q^2}\right)\sum_{k=0}^{K-1} M^{(k)} \leq &\left(\gamma^2\frac{4C^2}{(1-q)^2}\right)\sigma^2 K + \left(\gamma\frac{C^2}{(1-q)^2}\right)\sigma^2 \\
&+ \left(\gamma^2\frac{12C^2}{(1-q)^2}\right)\zeta^2 K + \left(\frac{\gamma 3C^2}{(1-q)^2}\right)\zeta^2 \\
&+ \left(\frac{C^2}{1-q^2} + \gamma\frac{2C^2}{(1-q)^2}\right)\frac{\sum_{i=1}^{n}\left\|x_i^{(0)}\right\|^2}{n} \\
&+ \left(\gamma^2\frac{12C^2}{(1-q)^2} + \gamma\frac{3C^2}{1-q^2}\right)\sum_{k=0}^{K-1}\mathbb{E}\left\|\nabla f\left(\sum_l^n w_l^{(k)}x_l^{(k)}\right)\right\|^2
\end{aligned}$$

(28)

Note that since $q \in (0,1)$ it holds that $\frac{1}{1-q^2} \le \frac{1}{(1-q)^2}$. Thus,

$$
\left(1 - \gamma^2 \frac{12L^2C^2}{(1-q)^2} - \gamma \frac{3L^2C^2}{(1-q)^2}\right) \sum_{k=0}^{K-1} M^{(k)} \le \left(\gamma^2 \frac{4C^2}{(1-q)^2}\right) \sigma^2 K + \left(\gamma \frac{C^2}{(1-q)^2}\right) \sigma^2
$$
$$
+ \left(\gamma^2 \frac{12C^2}{(1-q)^2}\right) \zeta^2 K + \left(\frac{\gamma 3 C^2}{(1-q)^2}\right) \zeta^2
$$
$$
+ \left(\frac{C^2}{(1-q)^2} + \gamma \frac{2C^2}{(1-q)^2}\right) \frac{\sum_{i=1}^n \left\| x_i^{(0)} \right\|^2}{n}
$$
$$
+ \left(\gamma^2 \frac{12C^2}{(1-q)^2} + \gamma \frac{3C^2}{(1-q)^2}\right) \sum_{k=0}^{K-1} \mathbb{E} \left\| \nabla f\left(\frac{1}{n}\frac{1}{n}\sum_l^n w_l^{(k)} x_l^{(k)}\right) \right\|^2
$$
$$
\tag{29}
$$

Dividing both sides with $D_2 = 1 - \gamma^2 \frac{12L^2C^2}{(1-q)^2} - \gamma \frac{3L^2C^2}{(1-q)^2}$ completes the proof.

**Lemma 7:** *Under the definition of our problem and the Assumptions 1-3 we have that:*

$(i)$ $\quad \mathbb{E}_{\xi_i^{(k)}} \left\| \frac{\sum_{i=1}^n w_i^{(k)} \nabla F_i\left(x_i^{(k)}; \xi_i^{(k)}\right)}{n} \right\|^2 = \mathbb{E}_{\xi_i^{(k)}} \left\| \frac{\sum_{i=1}^n w_i^{(k)} \left[\nabla F_i\left(x_i^{(k)}; \xi_i^{(k)}\right) - \nabla f_i\left(x_i^{(k)}\right)\right]}{n} \right\|^2$

$$
+ \mathbb{E}_{\xi_i^{(k)}} \left\| \frac{\sum_{i=1}^n w_i^{(k)} \left[\nabla f_i\left(x_i^{(k)}\right)\right]}{n} \right\|^2
$$

$(ii)$ $\quad \mathbb{E}_{\xi_i^{(k)}} \left\| \frac{\sum_{i=1}^n w_i^{(k)} \left[\nabla F_i\left(x_i^{(k)}; \xi_i^{(k)}\right) - \nabla f_i\left(x_i^{(k)}\right)\right]}{n} \right\|^2 \le \sigma^2$

*Proof for (i):*

$$
\mathbb{E}_{\xi_i^{(k)}} \left\| \frac{\sum_{i=1}^n w_i^{(k)} \nabla F_i\left(x_i^{(k)}; \xi_i^{(k)}\right)}{n} \right\|^2 = \mathbb{E}_{\xi_i^{(k)}} \left\| \frac{\sum_{i=1}^n w_i^{(k)} \left[\nabla F_i\left(x_i^{(k)}; \xi_i^{(k)}\right) - \nabla f_i\left(x_i^{(k)}\right)\right]}{n} + \frac{\sum_{i=1}^n w_i^{(k)} \nabla f_i\left(x_i^{(k)}\right)}{n} \right\|^2
$$

$$
= \mathbb{E}_{\xi_i^{(k)}} \left\| \frac{\sum_{i=1}^n w_i^{(k)} \left[\nabla F_i\left(x_i^{(k)}; \xi_i^{(k)}\right) - \nabla f_i\left(x_i^{(k)}\right)\right]}{n} \right\|^2
$$

$$
+ \mathbb{E}_{\xi_i^{(k)}} \left\| \frac{\sum_{i=1}^n w_i^{(k)} \nabla f_i\left(x_i^{(k)}\right)}{n} \right\|^2
$$

$$
+ 2 \left\langle \frac{\sum_{i=1}^n w_i^{(k)} \left[\mathbb{E}_{\xi_i^{(k)}} \nabla F_i\left(x_i^{(k)}; \xi_i^{(k)}\right) - \nabla f_i\left(x_i^{(k)}\right)\right]}{n}, \frac{\sum_{i=1}^n w_i^{(k)} \nabla f_i\left(x_i^{(k)}\right)}{n} \right\rangle
$$

$$
= \mathbb{E}_{\xi_i^{(k)}} \left\| \frac{\sum_{i=1}^n w_i^{(k)} \left[\nabla F_i\left(x_i^{(k)}; \xi_i^{(k)}\right) - \nabla f_i\left(x_i^{(k)}\right)\right]}{n} \right\|^2
$$

$$
+ \mathbb{E}_{\xi_i^{(k)}} \left\| \frac{\sum_{i=1}^n w_i^{(k)} \nabla f_i\left(x_i^{(k)}\right)}{n} \right\|^2
$$

$$
\tag{30}
$$

where in the last equality the inner product becomes zero from the fact that $\mathbb{E}_{\xi_i^{(k)}}\left[\nabla F_i\left(x_i^{(k)};\xi_i^{(k)}\right)\right] = \nabla f_i\left(x_i^{(k)}\right)$.

***Proof for (ii):***

$$
\begin{aligned}
&\mathbb{E}_{\xi_i^{(k)}}\left\|\frac{\sum_{i=1}^n w_i^{(k)}\nabla F_i\left(x_i^{(k)};\xi_i^{(k)}\right) - \sum_{i=1}^n w_i^{(k)}\nabla f_i\left(x_i^{(k)}\right)}{n}\right\|^2 \\
&= \frac{1}{n^2}\mathbb{E}_{\xi_i^{(k)}}\left\|\sum_{i=1}^n w_i^{(k)}\left[\nabla F_i\left(x_i^{(k)};\xi_i^{(k)}\right) - \nabla f_i\left(x_i^{(k)}\right)\right]\right\|^2 \\
&= \frac{1}{n^2}\sum_{i=1}^n \mathbb{E}_{\xi_i^{(k)}}\left[\left(w_i^{(k)}\right)^2\left\|\nabla F_i\left(x_i^{(k)};\xi_i^{(k)}\right) - \nabla f_i\left(x_i^{(k)}\right)\right\|^2\right] \\
&\quad + \frac{2}{n^2}\sum_{i\neq j}\langle\mathbb{E}_{\xi_i^{(k)}}\left[\left(w_i^{(k)}\right)^2\left[\nabla F_i\left(x_i^{(k)};\xi_i^{(k)}\right) - \nabla f_i\left(x_i^{(k)}\right)\right]\right], \\
&\qquad\qquad \mathbb{E}_{\xi_j^{(k)}}\left[\left(w_i^{(k)}\right)^2\left[\nabla F_j\left(x_j^{(k)};\xi_j^{(k)}\right) - \nabla f_j\left(x_j^{(k)}\right)\right]\right]\rangle \\
&= \frac{1}{n^2}\sum_{i=1}^n \mathbb{E}_{\xi_i^{(k)}}\left[\left(w_i^{(k)}\right)^2\left\|\nabla F_i\left(x_i^{(k)};\xi_i^{(k)}\right) - \nabla f_i\left(x_i^{(k)}\right)\right\|^2\right] \\
&\leq \frac{1}{n^2}\sum_{i=1}^n\left(w_i^{(k)}\right)^2\sigma^2 \leq \frac{\sigma^2}{n^2}\left(\sum_{i=1}^n w_i^{(k)}\right)^2\frac{\sigma^2}{n^2}n^2 = \sigma^2
\end{aligned}
\tag{31}
$$

**Lemma 8:** *Let Assumptions 1-3 hold and let $D_1 = \left[\frac{1}{2} - \frac{L^2}{2}\left(\frac{12\gamma^2 C^2 + 3\gamma C^2}{(1-q)^2 D_2}\right)\right]$ and $D_2 = 1 - \gamma^2\frac{12L^2 C^2}{(1-q)^2} - \gamma\frac{3L^2 C^2}{(1-q)^2}$*

***Proof:***

$$
f\left(\overline{x}^{(k+1)}\right) = f\left(\frac{\mathbf{X}^{(k+1)}\mathbf{1}_{\bar{n}}}{n}\right) = f\left(\frac{\mathbf{X}^{(k)}\left[\mathbf{P}^{(k)}\right]^\top\mathbf{1}_{\bar{n}} - \gamma\nabla F\left(\mathbf{X}^{(k)},\xi^{(k)}\right)\left[\mathbf{P}^{(k)}\right]^\top\mathbf{1}_{\bar{n}}}{n}\right)
$$

since $X^{(k+1)} = \left[X^{(k)} - \gamma\nabla F\left(\mathbf{X}^{(k)},\xi^{(k)}\right)\right]\left[P^{(k)}\right]^T$.

$\left[\mathbf{P}^{(k)}\right]^\top\mathbf{1}_{\bar{n}} = w^{(k)}$, where each element of $w$ is the column sum of $\mathbf{P}$. Since $\mathbf{P}$ is row-stochastic, that is, every row sum equals 1, hence the sum of all elements of $\mathbf{P}$ with $n$ rows equals $n$. Therefore, $\sum_i^n w^{(k)} = n$. For a column-stochastic $\mathbf{P}$, $w^{(k)} = \mathbf{1}_{\bar{n}}$ as in the formulation in Assran et al. (2019) and Koloskova et al. (2020), but we will be continuing our proof without assuming $w^{(k)} = \mathbf{1}_{\bar{n}}$, and show convergence.

$$
\begin{aligned}
f\left(\overline{x}^{(k+1)}\right) &= f\left(\frac{\mathbf{X}^{(k)}w^{(k)}}{n} - \frac{\gamma\nabla F\left(\mathbf{X}^{(k)},\xi^{(k)}\right)w^{(k)}}{n}\right) \\
&\overset{L-\text{smooth}}{\leq} f\left(\frac{\mathbf{X}^{(k)}w^{(k)}}{n}\right) - \gamma\left\langle\nabla f\left(\frac{\mathbf{X}^{(k)}w^{(k)}}{n}\right), \frac{\nabla F\left(\mathbf{X}^{(k)},\xi^{(k)}\right)w^{(k)}}{n}\right\rangle + \frac{L\gamma^2}{2}\left\|\frac{\nabla F\left(\mathbf{X}^{(k)},\xi^{(k)}\right)w^{(k)}}{n}\right\|^2
\end{aligned}
\tag{32}
$$

Here we use the $L-$smoothness inequality: $f(y) \leq f(x) + \nabla f(x)^T(y-x) + \frac{L}{2}\|y-x\|_2^2$.
Taking expectation on both sides conditioned on $\mathcal{F}_k$:

$$
\mathbb{E}\left[f\left(\frac{\mathbf{X}^{(k+1)}1_{\bar{n}}}{n}\right)\mid\mathcal{F}_k\right] \leq f\left(\frac{\mathbf{X}^{(k)}w^{(k)}}{n}\right) - \gamma\left\langle\nabla f\left(\frac{\mathbf{X}^{(k)}w^{(k)}}{n}\right), \frac{\nabla F\left(\mathbf{X}^{(k)}\right)w^{(k)}}{n}\right\rangle
$$

$$
+ \mathbb{E}\left[\frac{L\gamma^2}{2}\left\|\frac{\nabla F\left(\mathbf{X}^{(k)},\xi^{(k)}\right)w^{(k)}}{n}\right\|^2\mid\mathcal{F}_k\right]
$$

$$
\overset{\text{Lemma 7[i]}}{=} f\left(\frac{\mathbf{X}^{(k)}w^{(k)}}{n}\right) - \gamma\left\langle\nabla f\left(\frac{\mathbf{X}^{(k)}w^{(k)}}{n}\right), \frac{\nabla F\left(\mathbf{X}^{(k)}\right)w^{(k)}}{n}\right\rangle
$$

$$
+ \frac{L\gamma^2}{2}\mathbb{E}\left[\left\|\frac{\sum_{i=1}^n w_i^{(k)}\nabla F_i\left(x_i^{(k)};\xi_i^{(k)}\right) - \sum_{i=1}^n w_i^{(k)}\nabla f_i\left(x_i^{(k)}\right)}{n}\right\|^2\mid\mathcal{F}_k\right]
$$

$$
+ \frac{L\gamma^2}{2}\mathbb{E}\left[\left\|\frac{\sum_{i=1}^n w_i^{(k)}\nabla f_i\left(x_i^{(k)}\right)}{n}\right\|^2\mid\mathcal{F}_k\right]
$$

$$
\overset{\text{Lemma 7[ii]}}{=} f\left(\frac{\mathbf{X}^{(k)}w^{(k)}}{n}\right) - \gamma\left\langle\nabla f\left(\frac{\mathbf{X}^{(k)}w^{(k)}}{n}\right), \frac{\nabla F\left(\mathbf{X}^{(k)}\right)w^{(k)}}{n}\right\rangle
$$

$$
+ \frac{L\gamma^2\sigma^2}{2} + \frac{L\gamma^2}{2}\mathbb{E}\left[\left\|\frac{\sum_{i=1}^n w_i^{(k)}\nabla f_i\left(x_i^{(k)}\right)}{n}\right\|^2\mid\mathcal{F}_k\right]
$$

$$
= f\left(\frac{\mathbf{X}^{(k)}w^{(k)}}{n}\right) - \frac{\gamma}{2}\left\|\nabla f\left(\frac{\mathbf{X}^{(k)}w^{(k)}}{n}\right)\right\|^2 - \frac{\gamma}{2}\left\|\frac{\nabla F\left(\mathbf{X}^{(k)}\right)w^{(k)}}{n}\right\|^2,
$$

$$
+ \frac{\gamma}{2}\left\|\nabla f\left(\frac{\mathbf{X}^{(k)}w^{(k)}}{n}\right) - \frac{\nabla F\left(\mathbf{X}^{(k)}\right)w^{(k)}}{n}\right\|^2 + \frac{L\gamma^2\sigma^2}{2}
$$

$$
+ \frac{L\gamma^2}{2}\mathbb{E}\left[\left\|\frac{\sum_{i=1}^n w_i^{(k)}\nabla f_i\left(x_i^{(k)}\right)}{n}\right\|^2\mid\mathcal{F}_k\right]
$$

$$
\tag{33}
$$

where in the last step we expand the inner product using the expression $\|a-b\|^2 = \|a\|^2 + \|b\|^2 - 2\langle a,b\rangle$.

Now taking expectation with respect to $\mathcal{F}_k$ and using the tower property, we get,

$$
\mathbb{E}\left[f\left(\frac{\mathbf{X}^{(k+1)}1_{\bar{n}}}{n}\right)\right] \leq \mathbb{E}\left[f\left(\frac{\mathbf{X}^{(k)}w^{(k)}}{n}\right)\right] - \frac{\gamma}{2}\mathbb{E}\left[\left\|\nabla f\left(\frac{\mathbf{X}^{(k)}w^{(k)}}{n}\right)\right\|^2\right] - \frac{\gamma}{2}\mathbb{E}\left[\left\|\frac{\nabla F\left(\mathbf{X}^{(k)}\right)w^{(k)}}{n}\right\|^2\right],
$$

$$
+ \frac{\gamma}{2}\mathbb{E}\left[\left\|\nabla f\left(\frac{\mathbf{X}^{(k)}w^{(k)}}{n}\right) - \frac{\nabla F\left(\mathbf{X}^{(k)}\right)w^{(k)}}{n}\right\|^2\right] + \frac{L\gamma^2\sigma^2}{2}
$$

$$
+ \frac{L\gamma^2}{2}\mathbb{E}\left[\left\|\frac{\sum_{i=1}^n w_i^{(k)}\nabla f_i\left(x_i^{(k)}\right)}{n}\right\|^2\right]
$$

$$
= \mathbb{E}\left[f\left(\frac{\mathbf{X}^{(k)}w^{(k)}}{n}\right)\right] - \frac{\gamma}{2}\mathbb{E}\left[\left\|\nabla f\left(\frac{\mathbf{X}^{(k)}w^{(k)}}{n}\right)\right\|^2\right] - \frac{\gamma - L\gamma^2}{2}\mathbb{E}\left[\left\|\frac{\nabla F\left(\mathbf{X}^{(k)}\right)w^{(k)}}{n}\right\|^2\right]
$$

$$
+ \frac{\gamma}{2}\mathbb{E}\left[\left\|\nabla f\left(\frac{\mathbf{X}^{(k)}w^{(k)}}{n}\right) - \frac{\nabla F\left(\mathbf{X}^{(k)}\right)w^{(k)}}{n}\right\|^2\right] + \frac{L\gamma^2\sigma^2}{2}
$$

$$
\tag{34}
$$

Let us now focus on finding an upper bound for the quantity $\mathbb{E}\left[\left\|\nabla f\left(\frac{\mathbf{X}^{(k)}w^{(k)}}{n}\right) - \frac{\nabla F\left(\mathbf{X}^{(k)}\right)w^{(k)}}{n}\right\|^2\right]$

$$
\begin{aligned}
\mathbb{E}\left[\left\|\nabla f\left(\frac{\mathbf{X}^{(k)}w^{(k)}}{n}\right) - \frac{\nabla F\left(\mathbf{X}^{(k)}\right)w^{(k)}}{n}\right\|^2\right] &= \mathbb{E}\left[\left\|\nabla f(\frac{1}{n}\sum_l^n w_j^{(k)}x_j) - \frac{\sum_{i=1}^n w_i^{(k)}\nabla f_i\left(x_i^{(k)}\right)}{n}\right\|^2\right] \\
&= \mathbb{E}\left[\left\|\frac{1}{n}\sum_i^n \nabla f_i(\frac{1}{n}\sum_l^n w_l^{(k)}x_l) - \frac{\sum_{i=1}^n w_i^{(k)}\nabla f_i\left(x_i^{(k)}\right)}{n}\right\|^2\right] \\
&= \mathbb{E}\left[\left\|\frac{\sum_i^n \nabla f_i(\frac{1}{n}\sum_l^n w_l^{(k)}x_l) - \sum_{i=1}^n w_i^{(k)}\nabla f_i\left(x_i^{(k)}\right)}{n}\right\|^2\right] \\
&= \mathbb{E}\left[\left\|\frac{1}{n}\sum_i^n\left[\nabla f_i(\frac{1}{n}\sum_l^n w_l^{(k)}x_l) - w_i^{(k)}\nabla f_i\left(x_i^{(k)}\right)\right]\right\|^2\right] \\
&\overset{\text{Jensen}}{\leq} \frac{1}{n}\sum_i^n \mathbb{E}\left[\left\|\nabla f_i(\frac{1}{n}\sum_l^n w_l^{(k)}x_l) - w_i^{(k)}\nabla f_i\left(x_i^{(k)}\right)\right\|^2\right] \\
&\overset{\text{Cauchy-Schwartz}}{\leq} \frac{1}{n}\sum_i^n \mathbb{E}\left[\left[1 + max\left(w_i^{(k)}\right)^2\right]\left\|\nabla f_i(\frac{1}{n}\sum_l^n w_l^{(k)}x_l) - \nabla f_i\left(x_i^{(k)}\right)\right\|^2\right] \\
&\leq \frac{1+\frac{n^2}{4}}{n}\sum_i^n \mathbb{E}\left[\left\|\nabla f_i(\frac{1}{n}\sum_l^n w_l^{(k)}x_l) - \nabla f_i\left(x_i^{(k)}\right)\right\|^2\right] \\
&\overset{L-\text{ smooth}}{\leq} \frac{L^2\left(1+\frac{n^2}{4}\right)}{n}\sum_{i=1}^n \mathbb{E}\left[\left\|\frac{1}{n}\sum_l^n w_l^{(k)}x_l - x_i^{(k)}\right\|^2\right] \\
&= \frac{L_1^2}{n}\sum_{i=1}^n Q_i^{(k)}
\end{aligned}
\tag{35}
$$

where $L_1^2 = \frac{L^2\left(1+\frac{n^2}{4}\right)}{n}$. Thus we have,

$$
\begin{aligned}
\mathbb{E}\left[f\left(\frac{\mathbf{X}^{(k+1)}1_{\bar{n}}}{n}\right)\right] \leq{}& \mathbb{E}\left[f\left(\frac{\mathbf{X}^{(k)}w^{(k)}}{n}\right)\right] - \frac{\gamma}{2}\mathbb{E}\left[\left\|\nabla f\left(\frac{\mathbf{X}^{(k)}w^{(k)}}{n}\right)\right\|^2\right] - \frac{\gamma - L\gamma^2}{2}\mathbb{E}\left[\left\|\frac{\nabla F\left(\mathbf{X}^{(k)}\right)w^{(k)}}{n}\right\|^2\right] \\
&+ \frac{\gamma L_1^2}{2n}\sum_{i=1}^n Q_i^{(k)} + \frac{L\gamma^2\sigma^2}{2}
\end{aligned}
\tag{36}
$$

By rearranging,

$$
\begin{aligned}
\frac{\gamma}{2}\mathbb{E}\left[\left\|\nabla f\left(\frac{\mathbf{X}^{(k)}w^{(k)}}{n}\right)\right\|^2\right] + \frac{\gamma - L\gamma^2}{2}\mathbb{E}\left[\left\|\frac{\nabla F\left(\mathbf{X}^{(k)}\right)w^{(k)}}{n}\right\|^2\right] \leq{}& \mathbb{E}\left[f\left(\frac{\mathbf{X}^{(k)}w^{(k)}}{n}\right)\right] - \mathbb{E}\left[f\left(\frac{\mathbf{X}^{(k+1)}1_{\bar{n}}}{n}\right)\right] \\
&+ \frac{L\gamma^2\sigma^2}{2} + \frac{\gamma L_1^2}{2n}\sum_{i=1}^n Q_i^{(k)}
\end{aligned}
\tag{37}
$$

Let us now sum from $k = 0$ to $k = K - 1$:

$$
\frac{\gamma}{2} \sum_{k=0}^{K-1} \mathbb{E}\left[\left\|\nabla f\left(\frac{\mathbf{X}^{(k)} w^{(k)}}{n}\right)\right\|^2\right] + \frac{\gamma - L\gamma^2}{2} \sum_{k=0}^{K-1} \mathbb{E}\left[\left\|\frac{\nabla F\left(\mathbf{X}^{(k)}\right) w^{(k)}}{n}\right\|^2\right]
$$

$$
\leq \sum_{k=0}^{K-1}\left[\mathbb{E}\left[f\left(\frac{\mathbf{X}^{(k)} w^{(k)}}{n}\right)\right] - \mathbb{E}\left[f\left(\frac{\mathbf{X}^{(k+1)} w^{(k+1)}}{n}\right)\right]\right]
$$

$$
+ \sum_{k=0}^{K-1} \frac{L\gamma^2\sigma^2}{2} + \frac{\gamma L_1^2}{2n} \sum_{k=0}^{K-1} \sum_{i=1}^{n} Q_i^{(k)}
$$

$$
\leq \mathbb{E}\left[f\left(\frac{\mathbf{X}^{(0)} w^{(0)}}{n}\right)\right] - \mathbb{E}\left[f\left(\frac{\mathbf{X}^{(k)} w^{(k)}}{n}\right)\right]
$$

$$
\leq f\left(\frac{1}{n}\sum_{l}^{n} w_l^{(0)} x_l^{(0)}\right) - f^*
$$

$$
+ \frac{LK\gamma^2\sigma^2}{2} + \frac{\gamma L_1^2}{2} \sum_{k=0}^{K-1} \underbrace{\frac{1}{n}\sum_{i=1}^{n} Q_i^{(k)}}_{M_k}
$$

$$\tag{38}$$

The last inequality holds because $f^*$ is the theoretical global infimum of our problem.
Using the bound for the expression $\sum_{k=0}^{K-1} M_k$ from Lemma 6, we obtain:

$$
\frac{\gamma}{2} \sum_{k=0}^{K-1} \mathbb{E}\left[\left\|\nabla f\left(\frac{\mathbf{X}^{(k)} w^{(k)}}{n}\right)\right\|^2\right] + \frac{\gamma - L\gamma^2}{2} \sum_{k=0}^{K-1} \mathbb{E}\left[\left\|\frac{\nabla F\left(\mathbf{X}^{(k)}\right) w^{(k)}}{n}\right\|^2\right]
$$

$$
\leq f\left(\frac{1}{n}\sum_{l}^{n} w_l^{(0)} x_l^{(0)}\right) - f^* + \frac{LK\gamma^2\sigma^2}{2}
$$

$$
+ \frac{\gamma L^2}{2} \frac{4\gamma^2 C^2\sigma^2 K + \gamma C^2\sigma^2}{(1-q)^2 D_2} + \frac{\gamma L^2}{2} \frac{12\gamma^2 C^2\zeta^2 K + 3\gamma C^2\zeta^2}{(1-q)^2 D_2}
$$

$$
+ \frac{\gamma L^2}{2}\left(\frac{12\gamma^2 C^2 + 3\gamma C^2}{(1-q)^2 D_2}\right) \sum_{k=0}^{K} \mathbb{E}\left\|\nabla f\left(\sum_{l}^{n} w_l^{(k)} x_l^{(k)}\right)\right\|^2
$$

$$
+ \frac{\gamma L^2}{2}\left(\frac{C^2 + 2\gamma C^2}{(1-q)^2 D_2}\right) \frac{\sum_{i=1}^{n}\left\|x_i^{(0)}\right\|^2}{n}
$$

$$\tag{39}$$

By using $L < L_1$ and rearranging and dividing all terms by $\gamma K$, we obtain:

$$
\frac{1}{K}\left(\left[\frac{1}{2} - \frac{L_1^2}{2}\left(\frac{12\gamma^2 C^2 + 3\gamma C^2}{(1-q)^2 D_2}\right)\right] \sum_{k=0}^{K-1} \mathbb{E}\left\|\nabla f\left(\frac{1}{n}\sum_{l}^{n} w_l^{(k)} x_l^{(k)}\right)\right\|^2 + \frac{1 - L_1\gamma}{2} \sum_{k=0}^{K-1} \mathbb{E}\left\|\frac{\nabla F\left(\mathbf{X}^{(k)}\right) w^{(k)}}{n}\right\|^2\right)
$$

$$
\leq \frac{f\left(\frac{1}{n}\sum_{l}^{n} w_l^{(0)} x_l^{(0)}\right) - f^*}{\gamma K} + \frac{L_1\gamma\sigma^2}{2} + \frac{4L_1^2\gamma^2 C^2\sigma^2 + 12L_1^2\gamma^2 C^2\zeta^2}{2(1-q)^2 D_2} + \frac{\gamma L_1^2 C^2\sigma^2 + 3L_1^2\gamma C^2\zeta^2}{2K(1-q)^2 D_2}
$$

$$
+ \left(\frac{L_1^2 C^2 + 2L_1^2\gamma C^2}{2(1-q)^2 D_2 K}\right) \frac{\sum_{i=1}^{n}\left\|x_i^{(0)}\right\|^2}{n}
$$

$$\tag{40}$$

By defining $D_1 = \left[\frac{1}{2} - \frac{L_1^2}{2}\left(\frac{12\gamma^2 C^2 + 3\gamma C^2}{(1-q)^2 D_2}\right)\right]$, the proof is complete.

**Proof of Theorem 1:** Let $\gamma \le min\{\frac{(1-q)^2}{60L^2C^2}, 1\}$. Then,

$$D_2 = 1 - \frac{\gamma^2 12L_1^2C^2}{(1-q)^2} - \frac{\gamma 3L_1^2C^2}{(1-q)^2} \overset{(\gamma^2 < \gamma)}{\ge} 1 - \frac{\gamma 15L_1^2C^2}{(1-q)^2} \ge 1 - \frac{1}{4} \ge \frac{1}{2}$$

$$D_1 = \frac{1}{2} - \frac{L_1^2}{2}\left(\frac{12\gamma^2C^2 + 3\gamma C^2}{(1-q)^2 D_2}\right) \overset{(\gamma^2 < \gamma)}{\ge} \frac{1}{2} - \frac{15\gamma C^2 L_1^2}{2(1-q)^2 D_2} \ge \frac{1}{2} - \frac{1}{8D_2} \ge \frac{1}{4}$$

By substituting these bounds in Lemma 8, we obtain,

$$\frac{1}{4}\frac{\sum_{k=0}^{K-1}\mathbb{E}\left\|\nabla f\left(\frac{1}{n}\sum_{l=1}^{n}w_l^{(k)}x_l^{(k)}\right)\right\|^2}{K} \le \frac{1}{K}\left(\frac{1}{4}\sum_{k=0}^{K-1}\mathbb{E}\left\|\nabla f\left(\frac{1}{n}\sum_{l=1}^{n}w_l^{(k)}x_l^{(k)}\right)\right\|^2 + \frac{1-L_1\gamma}{2}\sum_{k=0}^{K-1}\mathbb{E}\left\|\frac{\nabla F\left(\mathbf{X}^{(k)}\right)w^{(k)}}{n}\right\|^2\right)$$

$$\le \frac{f\left(\frac{1}{n}\sum_l^n w_l^{(0)}x_l^{(0)}\right) - f^*}{\gamma K} + \frac{L_1\gamma\sigma^2}{2} + \frac{4L_1^2\gamma^2C^2\sigma^2 + 12L_1^2\gamma^2C^2\zeta^2}{(1-q)^2}$$

$$+ \frac{\gamma L_1^2C^2\sigma^2 + 3L_1^2\gamma C^2\zeta^2}{K(1-q)^2} + \left(\frac{L_1^2C^2 + 2L_1^2\gamma C^2}{(1-q)^2 K}\right)\frac{\sum_{i=1}^{n}\left\|x_i^{(0)}\right\|^2}{n}$$

$$(41)$$

Let us now substitute in the above expression $\gamma = \sqrt{n/K}$. This can be done due to the lower bound on the total number of iterations K where guarantees that $\sqrt{n/K} \le min\left\{\frac{(1-q)^2}{60L^2C^2}, 1\right\}$

$$\frac{1}{4}\frac{\sum_{k=0}^{K-1}\mathbb{E}\left\|\nabla f\left(\frac{1}{n}\sum_{l=1}^{n}w_l^{(k)}x_l^{(k)}\right)\right\|^2}{K} \le \frac{f\left(\frac{1}{n}\sum_l^n w_l^{(0)}x_l^{(0)}\right) - f^*}{\sqrt{nK}} + \sqrt{\frac{n}{K}}\frac{L_1\sigma^2}{2} + \frac{n}{K}\frac{4L_1^2C^2\sigma^2 + 12L_1^2C^2\zeta^2}{(1-q)^2}$$

$$+ \sqrt{\frac{n}{K}}\frac{L_1^2C^2\sigma^2 + 3L_1^2C^2\zeta^2}{K(1-q)^2} + \left(\frac{L_1^2C^2}{(1-q)^2 K}\right)\frac{\sum_{i=1}^{n}\left\|x_i^{(0)}\right\|^2}{n}$$

$$+ \sqrt{\frac{n}{K}}\left(\frac{2L_1^2\gamma C^2}{(1-q)^2 K}\right)\frac{\sum_{i=1}^{n}\left\|x_i^{(0)}\right\|^2}{n}$$

$$\le \frac{f\left(\frac{1}{n}\sum_l^n w_l^{(0)}x_l^{(0)}\right) - f^*}{\sqrt{nK}} + \frac{L_1^2C^2}{K(1-q)^2}\left[(4\sigma^2 + 12\zeta^2)n + \frac{\sum_{i=1}^{n}\left\|x_i^{(0)}\right\|^2}{n}\right]$$

$$+ \sqrt{\frac{n}{K}}\left[\frac{L_1^2C^2\sigma^2 + 3L_1^2C^2\zeta^2}{K(1-q)^2} + \left(\frac{2L_1^2\gamma C^2}{(1-q)^2 K}\right)\frac{\sum_{i=1}^{n}\left\|x_i^{(0)}\right\|^2}{n}\right]$$

$$(42)$$

Let $P_1 = \left[(4\sigma^2 + 12\zeta^2)n + \frac{\sum_{i=1}^{n}\left\|x_i^{(0)}\right\|^2}{n}\right]$

and $P_2 = \left[\frac{L_1^2C^2\sigma^2 + 3L_1^2C^2\zeta^2}{(1-q)^2} + \left(\frac{2L_1^2\gamma C^2}{(1-q)^2}\right)\frac{\sum_{i=1}^{n}\left\|x_i^{(0)}\right\|^2}{n}\right]$.

If

$$\frac{L_1^2C^2}{K(1-q)^2}P_1 \le \frac{f\left(\frac{1}{n}\sum_l^n w_l^{(0)}x_l^{(0)}\right) - f^*}{\sqrt{nK}},$$

$$\sqrt{\frac{n}{K}}\frac{P_2}{K} \le \frac{f\left(\frac{1}{n}\sum_l^n w_l^{(0)}x_l^{(0)}\right) - f^*}{\sqrt{nK}}$$

That is,

$$K \geq max \left\{ \frac{nL_1^4 C^4 (60)^2}{(1-q)^4}, \frac{nL_1^4 C^4 P_1}{(1-q)^2 \left( \frac{f\left(\frac{1}{n}\sum_l^n w_l^{(0)} x_l^{(0)}\right) - f^*}{\sqrt{nK}} \right)^2}, \frac{nP_2}{\frac{f\left(\frac{1}{n}\sum_l^n w_l^{(0)} x_l^{(0)}\right) - f^*}{\sqrt{nK}}} \right\} \quad (43)$$

Then

$$\frac{1}{4} \frac{\sum_{k=0}^{K-1} \mathbb{E} \left\| \nabla f \left( \frac{1}{n} \sum_{l=1}^n w_l^{(k)} x_l^{(k)} \right) \right\|^2}{K} \leq 3 \frac{f\left(\frac{1}{n}\sum_l^n w_l^{(0)} x_l^{(0)}\right) - f^*}{\sqrt{nK}} \quad (44)$$

**Proof of Theorem 2:** From Lemma 6 we have that:

$$\begin{aligned}
\frac{1}{K} \sum_{k=0}^{K-1} M^{(k)} \leq & \left( \gamma^2 \frac{4C^2}{(1-q)^2 D_2} \right) \sigma^2 + \left( \gamma \frac{C^2}{(1-q)^2 D_2} \right) \frac{\sigma^2}{K} \\
& + \left( \gamma^2 \frac{12C^2}{(1-q)^2 D_2} \right) \zeta^2 + \left( \frac{\gamma 3 C^2}{(1-q)^2 D_2} \right) \frac{\zeta^2}{K} \\
& + \left( \frac{C^2}{(1-q)^2 D_2 K} + \gamma \frac{2C^2}{(1-q)^2 D_2 K} \right) \frac{\sum_{i=1}^n \left\| \boldsymbol{x}_i^{(0)} \right\|^2}{n} \\
& + \left( \gamma^2 \frac{12C^2}{(1-q)^2 D_2} + \gamma \frac{3C^2}{(1-q)^2 D_2} \right) \frac{\sum_{k=0}^{K-1} \mathbb{E} \left\| \nabla f \left( \frac{1}{n} \sum_l^n w_l^{(k)} x_l^{(k)} \right) \right\|^2}{K}
\end{aligned} \quad (45)$$

Using the assumptions of Theorem 1 and step size $\gamma = \sqrt{n/K}$,

$$\frac{1}{K} \sum_{k=0}^{K-1} M^{(k)} \leq \mathcal{O} \left( \frac{1}{K} + \frac{1}{K\sqrt{K}} \right) \quad (46)$$

Now using the above upper bound in 46 and the result of Theorem 1, we obtain:

$$\begin{aligned}
\frac{1}{K} \sum_{k=0}^{K-1} \frac{1}{n} \sum_{i=1}^n \mathbb{E} \left\| \nabla f \left( \boldsymbol{x}_i^k \right) \right\|^2 &= \frac{1}{K} \sum_{k=0}^{K-1} \frac{1}{n} \sum_{i=1}^n \mathbb{E} \left\| \nabla f \left( \boldsymbol{x}_i^k \right) + \nabla f \left( \frac{1}{n} \sum_{l=1}^n w_l^{(k)} x_l^{(k)} \right) - \nabla f \left( \frac{1}{n} \sum_{l=1}^n w_l^{(k)} x_l^{(k)} \right) \right\|^2 \\
&\leq \frac{1}{K} \sum_{k=0}^{K-1} \frac{1}{n} \sum_{i=1}^n 2\mathbb{E} \left\| \nabla f \left( \boldsymbol{x}_i^k \right) - \nabla f \left( \frac{1}{n} \sum_{l=1}^n w_l^{(k)} x_l^{(k)} \right) \right\|^2 + 2\mathbb{E} \left\| \nabla f \left( \frac{1}{n} \sum_{l=1}^n w_l^{(k)} x_l^{(k)} \right) \right\|^2 \\
&= \frac{1}{K} \sum_{k=0}^{K-1} \frac{1}{n} \sum_{i=1}^n 2\mathbb{E} \left\| \nabla f \left( \boldsymbol{x}_i^k \right) - \nabla f \left( \frac{1}{n} \sum_{l=1}^n w_l^{(k)} x_l^{(k)} \right) \right\|^2 + \frac{1}{K} \sum_{k=0}^{K-1} \frac{1}{n} \sum_{i=1}^n 2\mathbb{E} \left\| \nabla f \left( \frac{1}{n} \sum_{l=1}^n w_l^{(k)} x_l^{(k)} \right) \right\|^2 \\
&= 2 \frac{1}{K} \sum_{k=0}^{K-1} \frac{1}{n} \sum_{i=1}^n \mathbb{E} \left\| \nabla f \left( \boldsymbol{x}_i^k \right) - \nabla f \left( \frac{1}{n} \sum_{l=1}^n w_l^{(k)} x_l^{(k)} \right) \right\|^2 + 2 \frac{1}{K} \sum_{k=0}^{K-1} \mathbb{E} \left\| \nabla f \left( \frac{1}{n} \sum_{l=1}^n w_l^{(k)} x_l^{(k)} \right) \right\|^2 \\
\overset{L-smooth}{=} & 2L^2 \frac{1}{K} \sum_{k=0}^{K-1} \frac{1}{n} \sum_{i=1}^n \mathbb{E} \left\| \boldsymbol{x}_i^k - \frac{1}{n} \sum_{l=1}^n w_l^{(k)} x_l^{(k)} \right\|^2 + 2 \frac{1}{K} \sum_{k=0}^{K-1} \mathbb{E} \left\| \nabla f \left( \frac{1}{n} \sum_{l=1}^n w_l^{(k)} x_l^{(k)} \right) \right\|^2 \\
\overset{\leq}{\underset{(46)+(44)}{}} & \mathcal{O} \left( \frac{1}{\sqrt{nK}} + \frac{1}{K} + \frac{1}{K^{3/2}} \right)
\end{aligned}$$

$$(47)$$

