# OpenReview forum: "P2PRISM - Peer to peer learning with individual prism for secure aggregation"
_ICLR.cc/2023/Conference — Submitted to ICLR 2023_

### Official Review · Reviewer_sRwk · 2022-10-25

**Confidence:** 2
**Correctness:** 2
**Technical Novelty And Significance:** 2
**Empirical Novelty And Significance:** 2
**Recommendation:** 3

**Clarity, Quality, Novelty And Reproducibility:**

Clarity is lacking in this paper as discussed in the previous weakness section. Though an interesting protocol is proposed.  The efficacy of the novelty could not be evaluated.

**Strength And Weaknesses:**

Strength: This paper considers a fairly interesting setting where no centralized trusted server is presented to aggregate the model updates as in federated learning settings. And it illustrates the possible malicious byzantine attacks that prior works cannot handle. Their protocol is clearly presented.

Weakness:
- The intuition behind the construction is not well presented. Especially the reward/penalize part.
- A big portion of technical novelty may go to the flip-score proposed by Sharma et al. (2021).
- They assume that their way of using flip-score should trim out the anomalies. This does not look straight to me. Neither does the later convergence analysis. It deserves a better explanation or formal proof.

Other thing:
- Running title says for ICLR 2022.

**Summary Of The Paper:**

This paper proposes a new secure aggregation protocol P2PRISM for decentralized peer-to-peer learning, against malicious adversaries, in various network topologies.

**Summary Of The Review:**

It might be better to emphasize the technical novelty of the proposed protocol and provide more intuition about why it's designed in this way. I think this paper needs improvement before acceptance.

---

> ### Author Response · Authors · 2022-11-09
> **Defense intuition, technical novelty, and convergence analysis**
>
> Thank you for the comments. Please find our response below.
>
> $Intuition$ -
>
> Relevance of flip-score - Given a benign reference direction of gradient updates, the metric of flip-score (FS) captures all untargeted model poisoning attacks, all of which aim to steer a model away from the optima. By construction (see Eqn (2)), a large FS indicates that either a large number of model parameters are made to flip direction, or a small number of them are flipped by a large magnitude, or both.
> However, it is not possible to set a hard threshold on FS above which the gradients can be said to be malicious, as the model goes through varying contours as it approaches the optima during training, and may occasionally require high FS moves to maneuver through the undulating  trajectory.
>
> At the same time, unlike [1], trimming out a fixed number of updates does not ensure byzantine resilience as P2PL can have a variable number of malicious nodes in every neighborhood. We therefore use an adaptive technique by suspecting updates that have sufficiently larger FS than the current median value of FS as observed by each node. We impose an upper cap on this median value to handle the case where a majority of malicious nodes poisons the median. The cutoff FS is affected by the defense parameter $\mu$, where $\mu=0$ is the most conservative approach that cuts off all updates with flip-score greater than the median, and higher $\mu$ values are more lenient.
>
> $Reputation$ $mechanism$ $intuition$ -
>
> Below the cutoff FS, every node is given a reward of 1 unit. The reward is not made variable to ensure 1) it is not easy for a node to accumulate high reputation and suddenly act malicious, 2) a bias is not created towards a few nodes with higher reputation.
>
> Penalty is however variable, and depends on how far a node’s FS is from the median value as compared to the distance between the median and the minimum value. Values closer to the median are likely to be false positives and are penalized less than those far away. The penalty is upper-capped to avoid over-penalization so that a node always has a chance to redeem itself by acting benign in the subsequent iterations.
>
> In every round, the reputation values are normalized to ensure that a node cannot accumulate reputation, and take advantage of it to suddenly act malicious. The normalization factor includes only the positive reputation values so as to prevent the denominator from attaining a very low value if there are sufficient nodes with negative reputation. We avoid taking the absolute value of negative reputation values in this calculation to prevent over-attenuation of reputation score so that the past values still remain relevant.
>
> $Technical$ $novelty$ -
>
> The state-of-the-art attack uses the idea of gradient ascent to poison the models. Use of FS ensures that all gradient ascent moves are flagged as malicious. However, it is not trivial to extend the concept of FS from FL to P2PL. In FL, an upper limit on the number of malicious clients $c_{max}$ can be assumed, and a fixed number of updates are flagged in every round. The penalty and reward values are also functions of this $c_{max}$. This is not valid in P2PL where the number of malicious nodes can be different in different neighborhoods. Our method does not depend on any hard threshold on FS or $c_{max}$, but on the current training dynamics as observed by every node, by tying the defense policy to the median FS value, and constructing a penalty-reward mechanism that takes care of all nuances in P2PL (as described above) without assuming any constant $c_{max}$. It is also capable of handling local majority of malicious nodes while keeping the consensus distance under control.
>
> ***Convergence analysis*** -
>
> The very design of FS identifies gradient ascent attacks which covers *all* existing untargeted model poisoning attacks (also described above). The parameter $\mu$ determines the strictness of the defense policy, with a low values close to 0 corresponding to a harsh conservative approach trimming out all anomalies. This is also empirically shown in our experiments. Our defense policy with the reputation mechanism effectively leads to a graph with changing topologies. In every round, a node is connected to a neighbor if it has a positive reputation, otherwise the connection is effectively dropped. The convergence analysis hence follows from [2] that already proves convergence with changing topologies. For further clarity, we will be including the entire analysis in our draft revision.
>
> [1]: Sharma, Atul, Wei Chen, Joshua Zhao, Qiang Qiu, Somali Chaterji, and Saurabh Bagchi. "TESSERACT: Gradient Flip Score to Secure Federated Learning Against Model Poisoning Attacks." arXiv preprint arXiv:2110.10108 (2021).
>
> [2]: Assran, Mahmoud, Nicolas Loizou, Nicolas Ballas, and Mike Rabbat. "Stochastic gradient push for distributed deep learning." In International Conference on Machine Learning, pp. 344-353. PMLR, 2019.

---

> ### Author Response · Authors · 2022-11-19
> **Complete convergence proof included in the main paper**
>
> We have now included the complete proof of convergence analysis of $P2PRISM$ in the Appendix section and have updated the paper addressing the comments as best as we can. The proof mainly describes how existing analysis in [1] is modified when the mixing matrix is no longer column stochastic but is row stochastic.
>
> [1] - Assran, Mahmoud, Nicolas Loizou, Nicolas Ballas, and Mike Rabbat. "Stochastic gradient push for distributed deep learning." In International Conference on Machine Learning, pp. 344-353. PMLR, 2019.

---

### Official Review · Reviewer_73SE · 2022-10-26

**Confidence:** 4
**Correctness:** 4
**Technical Novelty And Significance:** 3
**Empirical Novelty And Significance:** 3
**Recommendation:** 6

**Clarity, Quality, Novelty And Reproducibility:**

-the paper is in general well written, though references to FL seems more in the beginning than the intended contribution
-the paper introduces flip score computation to identify malicious nodes
-the paper shows experimental details that help reproducibility


**Strength And Weaknesses:**

strengths
-the paper proposes P2PRISM, a defense against malicious attacks in P2PL environment
-based on the direction of updates, the approach uses its own local update as the reference to detect parameter-wise direction flip in the updates received from its neighbors to identify malicious ones
-experiments are detailed and extensive

weakness
-there are many parameters used
-the reputation update mechanism seems naïve and could be compromised



**Summary Of The Paper:**

The paper describes the effect of malicious nodes in P2PL environment. It proposes a defense mechanism P2PRISM for reviving the nodes from the attack. Experiments are detailed and extensive and showcase the advantage of the proposed work.

**Summary Of The Review:**

The paper shows adequate contribution is addressing an important problem of malicious nodes in P2PL environments. The approach has been extensively evaluated to verify the paper’s claims

---

> ### Author Response · Authors · 2022-11-09
> **Defense parameters, Reputation mechanism**
>
> Thank you for the comments. We would like to clarify any confusions about the defense parameters and the reputation mechanism here.
>
> $Defense$ $parameters$ -
>
> The only parameter P2PRISM uses is $\mu$. This parameter can be interpreted in the following manner. The FS values observed by a node forms a certain distribution. $\mu$ decides how far away on the right of the median of this distribution, will the values be permissible. $\mu=0$ means that we flag all updates with FS greater than the median of this distribution to be malicious. $\mu=1$ means that we permit FS values as much to the right of the median, as the minimum FS is to the left of the median. Higher $\mu$ values allow even more skewed distributions, that is, updates with high FS are also allowed to contribute to the aggregation at a node. The choice of $\mu$ determines the strictness of the defense policy, a low value being harsh and conservative and a large value being lenient, allowing diverse updates to contribute to aggregation but at the risk of compromising security. It is left to the user to decide the value of $\mu$ depending upon the severity of the threat in a given training environment.
>
> $Reputation$ $mechanism$ -
>
> We describe the intuition behind the policies above for constructing the reputation mechanism in response to Reviewer sRwk. In short, it takes care of the following cases - the reward is constant to avoid creating bias towards any node, penalty is variable in proportion to the severity of the suspected attack, but avoids over and under penalization, redemption after penalization is made possible but difficult, and would take a node a few rounds of benign behavior, normalization of reputation ensures a node cannot accumulate reputation by acting benign and suddenly attack. We have also evaluated our system against the most powerful whitebox attacker that tries to adaptively break the defense by knowing the details of the defense mechanism. We would be interested in knowing any other corner cases in which the above scheme can be compromised so that we can work to make it even more robust.

---

### Official Review · Reviewer_FPrf · 2022-10-26

**Confidence:** 3
**Correctness:** 4
**Technical Novelty And Significance:** 3
**Empirical Novelty And Significance:** 3
**Recommendation:** 5

**Clarity, Quality, Novelty And Reproducibility:**

The algorithm itself is novel, but the general idea of using local updates for validation and Byzantine tolerance is not that novel.

**Strength And Weaknesses:**

Strength:
1. The paper provides an insightful discussion about the difference between Byzantine-tolerant machine learning in FL and P2PL.
2. This paper proposes P2PRISM, a novel secure aggregation protocol for Byzantine-tolerant P2PL.
3. Theoretical analysis for the convergence of P2PRISM is provided.
4. Empirical results show the the proposed method outperforms the basline

Weakness:
1. I would recommend using larger CV datasets such as cifar-10 instead of mnist, but it is not a bug issue.
2. The general idea of using using local updates for validation and Byzantine tolerance is actually not that novel. For example, the following 2 papers (zeno/zeno++ for short):
Xie, C., Koyejo, O., & Gupta, I. (2019). Zeno: Distributed Stochastic Gradient Descent with Suspicion-based Fault-tolerance. ICML. https://proceedings.mlr.press/v97/xie19b.html
Xie, C., Koyejo, O., & Gupta, I. (2020). Zeno++: Robust Fully Asynchronous SGD. ICML.  https://proceedings.mlr.press/v119/xie20c.html
Especially for zeno++, it uses the inner-product between the update on the validation server and the candidate update to calculate a score for Byzantine tolerance, which sounds very similar to the general idea of P2PRISM, though the formula of the scores are different.
Although zeno/zeno++ are originally for centralized learning, I think it's trivial to apply them to P2PL. Basically you just need to treat each worker as a server that uses the local training data to compute the validation score for zeno/zeno++.
I think that zeno/zeno++ should be a better baseline compared to trimmed mean, since zeno/zeno++ are the techniques closet to P2PRISM.
I highly recommend to at least give a detailed discussion about zeno/zeno++ in this paper. Experiments using modified zeno/zeno++ in P2PL as the baselines are also recommended.

**Summary Of The Paper:**

This paper proposes P2PRISM, a novel secure aggregation protocol for P2PL, with theoretical analysis for convergence and good empirical results.

**Summary Of The Review:**

This paper proposes P2PRISM, a novel secure aggregation protocol for P2PL, with theoretical analysis for convergence and good empirical results. Some important baselines are missing.

---

> ### Author Response · Authors · 2022-11-09
> **FL baselines in P2PL and datasets used in P2PRISM**
>
> Thank you for recommending these baselines. The papers were an interesting read. We would like to highlight that the idea of local validation is not novel in our work. It is just the nature of P2PL that it allows local validation that we have taken advantage of, that has not been highlighted in any previous work. The novelty lies in the design of the defense mechanism when we can no longer assume that in any neighborhood, the malicious nodes are in a minority. The novelty also lies in the analysis of P2PL, highlighting how it behaves differently from FL, and that the extension of techniques from FL that is non-trivial.
>
> Zeno assumes an upper limit $b$ on the number of malicious nodes. It trims out the $b$ updates that are flagged as malicious. This approach cannot work in P2PL because the limit $b$ can change in different neighborhoods, and is unknown. Choosing $b=m-1$ in the most conservative approach would lead to individual training and no benefit could be extracted from any collaboration. This applies to almost all the FL baselines, including Trimmed Mean, which is why they cannot be directly extended to P2PL. Zeno++ gets rid of the constant upper limit $b$ but uses a hard threshold on the zeno score computed which is hard to set as the model moves through varying undulating contours during training, even more in P2PL than in FL, that needs to be accounted for in an adaptive manner to minimize false positives and false negatives. P2PRISM handles this by tying the defense policy to the median FS, instead of assuming any constant upper limit on the number of malicious nodes in a neighborhood, or any hard threshold on the flip-score metric. Further, the reputation mechanism takes care of all the corner cases that could arise in the P2PL setting. We describe this in detail in response to Reviewer sRwk.
>
> Datasets used - We would also like to highlight that we have performed experiments not only on MNIST, but also on Fashion-MNIST, and Shakespeare (a text dataset), all on two different topologies - k-regular and power law graphs.

---

> ### Author Response · Authors · 2022-12-02
> **Comparison with Zeno**
>
> We have experimentally verified the arguments we made about Zeno in our last response. We compare P2PRISM with Zeno on the MNIST dataset with $m=128$ nodes out of which $c=16$ are malicious in a $k$-regular graph with $k=8,16,32$. We set the parameter $b$ in Zeno as $b = (c/m)*k$. Figure 1 (https://ibb.co/wcScMTC) shows the spread of test accuracy across the nodes as a function of time. We observe that Zeno fails to secure most of the benign nodes against the attack. In most cases, only the nodes that are surrounded by $\leq b$ malicious nodes are saved. This is even more clear in Figure 2 (https://ibb.co/r50Gr2G), where we show the final test accuracy of the nodes, with the benign nodes shown in green and the malicious in red. Recall that we have simulated a $k$-regular graph in which all nodes can be assumed to sit along a ring and communicate with its closest $k-1$ neighbors.
>
> Having shown the comparison of Zeno with P2PRISM for $c=16$, where Zeno fails to secure the nodes and P2PRISM observes complete success, we also show in the two figures, the performance of Zeno for lower $c$ values. We see that Zeno fails to secure the network of nodes even for $c=4,8$. This is particularly visible when $k$ is high ($k=32$). Since the nodes with $\geq b$ malicious neighbors are not protected, they not only get infected but spread the infection to all of their neighbors. When $k$ is high, the high connectivity spreads the infection fast and the entire network suffers.
>
> We hope that this experiment clarifies all questions that the reviewer had, and we thank them to recommend such a useful paper to us.
>
> To summarize, Zeno, or any other defense technique that relies on a fixed upper bound on the number of malicious nodes tend to fail in the P2P setting where the density of malicious nodes varies across regions. Hence, the performance of all these techniques is similar, here Zeno and Trimmed Mean.

---

### Author Response · Authors · 2022-11-19
**Paper revision**

Hello everyone. Thank you for all your reviews and suggestions. We have addressed the comments as best as we can, and have also made the updates in the latest paper revision. Since the rebuttal window is drawing to a close, we would like to see if there are any other concerns and questions that we can address. Thank you.

---

### Decision · Program_Chairs · 2023-01-20

**Decision:**

Reject

**Justification For Why Not Higher Score:**

Please see above.

**Justification For Why Not Lower Score:**

NA

**Metareview: Summary, Strengths And Weaknesses:**

 The paper proposes P2PRISM, a novel secure aggregation protocol for P2PL. The authors indicate vulnerabilities under malicious attacks in the P2PL setting,  as well as a provably robust algorithm. Empirical results show the the proposed method outperforms the (non-robust) baseline.

The paper could be better positioned w.r.t. related, and novelty should be discussed in comparison with papers brought up by the reviewers; the experimental comparison added certainly looks promising. The paper could also use polishing, along the lines indicated in the reviews.

**Summary Of Ac-Reviewer Meeting:**

NA